# FairBatch:
# Batch Selection for Model Fairness

**Yuji Roh**[1], **Kangwook Lee**[2], **Steven Euijong Whang**[*1], **Changho Suh**[1]
[1]KAIST, {yuji.roh,swhang,chsuh}@kaist.ac.kr
[2]University of Wisconsin-Madison, kangwook.lee@wisc.edu

## Abstract

Training a fair machine learning model is essential to prevent demographic disparity. Existing techniques for improving model fairness require broad changes in either data preprocessing or model training, rendering themselves difficult-to-adopt for potentially already complex machine learning systems. We address this problem via the lens of *bilevel optimization*. While keeping the standard training algorithm as an inner optimizer, we incorporate an outer optimizer so as to equip the inner problem with an additional functionality: *Adaptively selecting minibatch sizes for the purpose of improving model fairness*. Our batch selection algorithm, which we call FairBatch, implements this optimization and supports prominent fairness measures: equal opportunity, equalized odds, and demographic parity. FairBatch comes with a significant implementation benefit – it does not require any modification to data preprocessing or model training. For instance, a single-line change of PyTorch code for replacing batch selection part of model training suffices to employ FairBatch. Our experiments conducted both on synthetic and benchmark real data demonstrate that FairBatch can provide such functionalities while achieving comparable (or even greater) performances against the state of the arts. Furthermore, FairBatch can readily improve fairness of any pre-trained model simply via fine-tuning. It is also compatible with existing batch selection techniques intended for different purposes, such as faster convergence, thus gracefully achieving multiple purposes.

## 1 Introduction

Model fairness is becoming essential in a wide variety of machine learning applications. Fairness issues often arise in sensitive applications like healthcare and finance where a trained model must not discriminate among different individuals based on age, gender, or race.

While many fairness techniques have recently been proposed, they require a range of changes in either data generation or algorithmic design. There are two popular fairness approaches: (i) pre-processing where training data is debiased (Choi et al., 2020) or re-weighted (Jiang and Nachum, 2020), and (ii) in-processing in which an interested model is retrained via several fairness approaches such as fairness objectives (Zafar et al., 2017a;b), adversarial training (Zhang et al., 2018), or boosting (Iosifidis and Ntoutsi, 2019); see more related works discussed in depth in Sec. 5. However, these approaches may require nontrivial re-configurations in modern machine learning systems, which often consist of many complex components.

In an effort to enable easier-to-reconfigure implementation for fair machine learning, we address the problem via the lens of bilevel optimization where one problem is embedded within another. While keeping the standard training algorithm as the inner optimizer, we design an outer optimizer that equips the inner problem with an added functionality of improving fairness through batch selection.

Our main contribution is to develop a batch selection algorithm (called FairBatch) that implements this optimization via adjusting the batch sizes w.r.t. sensitive groups based on the fairness measure of an intermediate model, measured in the current epoch. For example, consider a task of predicting whether individual criminals re-offend in the future subject to satisfying equalized odds (Hardt et al., 2016) where the model accuracies must be the same across sensitive groups. In case the model is less

---

*Corresponding author

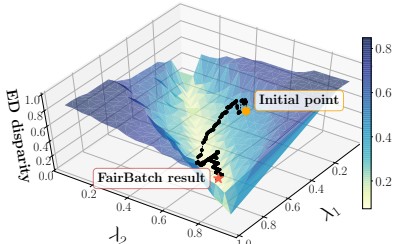

```
fairsampler = FairBatch(model, criterion, train_data,
                        batch_size, alpha, target_fairness)
loader = DataLoader(train_data, sampler = fairsampler)

for epoch in range(epochs):
  for i, data in enumerate(loader):
    # get the inputs; data is a list of [inputs, labels]
    inputs, labels = data

    ... model forward, backward, and optimization ...
```

(a) Accuracy difference across sensitive groups in the sense of equalized odds (that we denote as "ED disparity") when running FairBatch on the ProPublica COMPAS dataset.

(b) PyTorch code for model training where the batch selection is replaced with FairBatch.

Figure 1: The black path in the left figure shows how FairBatch adjusts the batch-size ratios of sensitive groups using two reweighting parameters $\lambda_1$ and $\lambda_2$ (hyperparameters employed in our framework to be described in Sec. 2), thus minimizing their ED disparity, i.e., achieving equalized odds. The code in the right figure shows how easily FairBatch can be incorporated in a PyTorch machine learning pipeline. It requires a single-line change to replace the existing sampler with FairBatch, marked in blue.

accurate for a certain group, FairBatch increases the batch-size ratio of that group in the next batch – see Sec. 3 for our adjusting mechanism described in detail. Fig. 1a shows FairBatch's behavior when running on the ProPublica COMPAS dataset (Angwin et al., 2016). For equalized odds, our framework (to be described in Sec. 2) introduces two reweighting parameters $(\lambda_1, \lambda_2)$ for the purpose of adjusting the batch-size ratios of two sensitive groups (in this experiment, men and women). After a few epochs, FairBatch indeed achieves *e*qualized *o*dds, i.e., the accuracy *disparity* between sensitive groups conditioned on the true label (denoted as "ED disparity") is minimized. FairBatch also supports other prominent group fairness measures: equal opportunity (Hardt et al., 2016) and demographic parity (Feldman et al., 2015).

A key feature of FairBatch is in its great usability and simplicity. It only requires a slight modification in the batch selection part of model training as demonstrated in Fig. 1b and does not require any other changes in data preprocessing or model training. Experiments conducted both on synthetic and benchmark real datasets (ProPublica COMPAS (Angwin et al., 2016), AdultCensus (Kohavi, 1996), and UTKFace (Zhang et al., 2017)) show that FairBatch exhibits greater (at least comparable) performances relative to the state of the arts (both spanning pre-processing (Kamiran and Calders, 2011; Jiang and Nachum, 2020) and in-processing (Zafar et al., 2017a;b; Zhang et al., 2018; Iosifidis and Ntoutsi, 2019) techniques) w.r.t. all aspects in consideration: accuracy, fairness, and runtime. In addition, FairBatch can improve fairness of any pre-trained model via fine-tuning. For example, Sec. 4.2 shows how FairBatch reduces the ED disparities of ResNet18 (He et al., 2016) and GoogLeNet (Szegedy et al., 2015) pre-trained models. Finally, FairBatch can be gracefully merged with other batch selection techniques typically used for faster convergence, thereby improving fairness faster as well.

**Notation** Let $w$ be the parameter of an interested classifier. Let $x \in \mathbb{X}$ be an input feature to the classifier, and let $\hat{y} \in \mathbb{Y}$ be the predicted class. Note that $\hat{y}$ is a function of $(x, w)$. We consider group fairness that intends to ensure fairness across distinct sensitive groups (e.g., men versus women). Let $z \in \mathbb{Z}$ be a sensitive attribute (e.g., gender). Consider the 0/1 loss: $\ell(y, \hat{y}) = \mathbb{1}(y \neq \hat{y})$, and let $m$ be the total number of train samples. Let $L_{y,z}(w)$ be the empirical risk aggregated over samples subject to $y = y$ and $z = z$: $L_{y,z}(w) := \frac{1}{m_{y,z}} \sum_{i:y_i=y,z_i=z} \ell(y_i, \hat{y}_i)$ where $m_{y,z} := |\{i : y_i = y, z_i = z\}|$. Similarly, we define $L_{y,\star}(w) := \frac{1}{m_{y,\star}} \sum_{i:y_i=y} \ell(y_i, \hat{y}_i)$ and $L_{\star,z}(w) := \frac{1}{m_{\star,z}} \sum_{i:z_i=z} \ell(y_i, \hat{y}_i)$ where $m_{y,\star} := |\{i : y_i = y\}|$ and $m_{\star,z} := |\{i : z_i = z\}|$. The overall empirical risk is written as $L(w) = \frac{1}{m} \sum_i \ell(y_i, \hat{y}_i)$. We utilize $\nabla$ for gradient and $\partial$ for subdifferential.

## 2 BILEVEL OPTIMIZATION FOR FAIRNESS

In order to systematically design an adaptive batch selection algorithm, we formalize an implicit connection between adaptive batch selection and bilevel optimization. Bilevel optimization consists of an outer optimization problem and an inner optimization problem. The inner optimizer solves an

---

**Algorithm 1:** Bilevel optimization with `MinibatchSGD`

---

Minibatch sampling distribution ← Uniform sampling
**for** *each epoch* **do**
    Draw minibatches according to minibatch sampling distribution
    **for** *each minibatch* **do**
        $\boldsymbol{w} \leftarrow$ `MinibatchSGD`$(\boldsymbol{w}, \text{each minibatch})$
    Update minibatch sampling distribution

---

inner optimization problem, and the outer optimizer solves an outer optimization problem based on the outcomes of inner optimization. By viewing the standard training algorithm such as stochastic gradient descent (SGD) (Bottou, 2010) as an inner optimizer and viewing the batch selection algorithm as an outer optimizer, the process of training a fair classifier can be seen as a process of solving a bilevel optimization problem.

**Batch selection + minibatch SGD = bilevel optimization solver** Consider a scenario where one is minimizing the overall empirical risk $L(\boldsymbol{w})$ via minibatch SGD. The minibatch SGD algorithm picks $b$ of the $m$ indices uniformly at random, say $j_1, j_2, \ldots, j_b$, and updates its iterate with $\frac{1}{b} \sum_{i=1}^{b} \nabla \ell(\mathrm{y}_{j_i}, \hat{\mathrm{y}}_{j_i})$, called a batch gradient. Note that a batch gradient is an unbiased estimate of the true gradient $\nabla L(\boldsymbol{w})$.

Since the empirical risk minimization (ERM) formulation does not take a fairness criterion into account, its minimizer usually does not satisfy the desired fairness criterion. To address this limitation of ERM, we adjust the way minibatches are drawn so that the desired fairness guarantee is satisfied. For instance, as we described in the introduction, we can draw minibatches with a larger number of train samples from a certain sensitive group so as to achieve a higher accuracy w.r.t. the group.

Once the minibatch distribution deviates from the uniform distribution, the batch gradient estimate is not anymore an unbiased gradient estimate of the overall empirical risk. Instead, it is an unbiased estimate of a *reweighted empirical risk*. In other words, if we draw train example $i$ with probability $p_i$ for all $i$ such that $\sum p_i = 1$, the batch gradient is an unbiased estimate of $L'(\boldsymbol{w}) = \sum_i p_i \ell(\mathrm{y}_i, \hat{\mathrm{y}}_i)$.

This observation enables us the following bilevel optimization-based interpretation of how batch selection interacts with inner optimization algorithm. At initialization, minibatch SGD optimizes the (unweighted) empirical risk. Based on the outcome of the inner optimization, the outer optimizer refines $\boldsymbol{p} := (p_1, p_2, \ldots, p_m)$, the sampling probability of each train example. The inner optimizer now takes minibatches drawn from a new distribution and reoptimizes the inner objective function. Due to the new minibatch distribution, the inner objective now becomes a reweighted empirical risk w.r.t. $\boldsymbol{p}$. This procedure is repeated until convergence. See Algorithm 1 for pseudocode.

Therefore, a batch selection algorithm together with an inner optimization algorithm can be viewed as a pair of outer optimizer and inner optimizer for the following bilevel optimization problem:

$$\min_{\boldsymbol{p}} \text{Cost}(\boldsymbol{w_p}), \; \boldsymbol{w_p} = \arg\min_{\boldsymbol{w}} L'(\boldsymbol{w}),$$

where $\text{Cost}(\cdot)$ captures the goal of the optimization.

Two questions arise. First, how can we design the cost function to capture a desired fairness criterion? Second, how can we design an update rule for the outer optimizer? Can we develop an algorithm with a provable guarantee? In the rest of this section, we show how one can design proper cost functions to capture various fairness criteria. In Sec. 3, we will develop an efficient update rule of FairBatch.

**Equal opportunity** For illustrative purpose, assume for now the binary setting ($\mathbb{Y} = \mathbb{Z} = \{0, 1\}$). A model satisfies equal opportunity (Hardt et al., 2016) if we have equal positive prediction rates conditioned on $\mathrm{y} = 1$, i.e., $L_{1,0}(\boldsymbol{w}) = L_{1,1}(\boldsymbol{w})$. Since the ERM formulation does not take the fairness criterion into account, these two quantities differ in general. To mitigate this, we adjust the sampling probability between $L_{1,0}(\boldsymbol{w})$ and $L_{1,1}(\boldsymbol{w})$. More specifically, we propose the following procedure to draw a sample. First, we randomly pick which subset of data to sample data from. We pick the set $\mathrm{y} = 1, \mathrm{z} = 0$ with probability $\lambda$, the set $\mathrm{y} = 1, \mathrm{z} = 1$ with probability $\frac{m_{1,\star}}{m} - \lambda$, and the set $\mathrm{y} = 0$ with probability $\frac{m_{0,\star}}{m}$. We then pick a sample from the chosen set, uniformly at random.

This leaves us with a single-dimensional outer optimization variable $\lambda$, which controls the sampling bias between data with $\mathrm{y} = 1, \mathrm{z} = 0$ and data with $\mathrm{y} = 1, \mathrm{z} = 1$. Also, we design the cost function as $|L_{1,0}(\boldsymbol{w}_\lambda) - L_{1,1}(\boldsymbol{w}_\lambda)|$ to capture the equal opportunity criterion. Thus, we have the following bilevel optimization problem:

$$\min_{\lambda \in [0, \frac{m_{1,\star}}{m}]} |L_{1,0}(\boldsymbol{w}_\lambda) - L_{1,1}(\boldsymbol{w}_\lambda)|, \ \boldsymbol{w}_\lambda = \arg\min_{\boldsymbol{w}} \lambda L_{1,0}(\boldsymbol{w}) + (\tfrac{m_{1,\star}}{m} - \lambda)L_{1,1}(\boldsymbol{w}) + \tfrac{m_{0,\star}}{m}L_{0,\star}(\boldsymbol{w}).$$

**Equalized odds** Similarly, we can design a bilevel optimization problem to capture equalized odds (Hardt et al., 2016), which desires the prediction to be independent from the sensitive attribute conditional on the true label, i.e., $L_{0,0}(\boldsymbol{w}) = L_{0,1}(\boldsymbol{w})$ and $L_{1,0}(\boldsymbol{w}) = L_{1,1}(\boldsymbol{w})$. Again, the empirical risk minimizer does not satisfy these two conditions in general. To mitigate these disparities, we adjust (i) the sampling probability between $L_{0,0}(\boldsymbol{w})$ and $L_{0,1}(\boldsymbol{w})$ and (ii) the sampling probability between $L_{1,0}(\boldsymbol{w})$ and $L_{1,1}(\boldsymbol{w})$. To achieve this, we use the following procedure to draw a sample. First, we pick the set $\mathrm{y} = 0, \mathrm{z} = 0$ with probability $\lambda_1$, the set $\mathrm{y} = 0, \mathrm{z} = 1$ with probability $\frac{m_{0,\star}}{m} - \lambda_1$, the set $\mathrm{y} = 1, \mathrm{z} = 0$ with probability $\lambda_2$, and the set $\mathrm{y} = 1, \mathrm{z} = 1$ with probability $\frac{m_{1,\star}}{m} - \lambda_2$. We then pick one data point at random from the chosen set. This leaves us with a two-dimensional outer optimization variable $\boldsymbol{\lambda} = (\lambda_1, \lambda_2)$. To capture the equalized odds criterion, we design the outer objective function as: $\max\{|L_{0,0}(\boldsymbol{w}) - L_{0,1}(\boldsymbol{w})|, |L_{1,0}(\boldsymbol{w}) - L_{1,1}(\boldsymbol{w})|\}$. This gives us the following bilevel optimization problem:

$$\min_{\boldsymbol{\lambda} \in [0, \frac{m_{0,\star}}{m}] \times [0, \frac{m_{1,\star}}{m}]} \max\{|L_{0,0}(\boldsymbol{w}_{\boldsymbol{\lambda}}) - L_{0,1}(\boldsymbol{w}_{\boldsymbol{\lambda}})|, |L_{1,0}(\boldsymbol{w}_{\boldsymbol{\lambda}}) - L_{1,1}(\boldsymbol{w}_{\boldsymbol{\lambda}})|\},$$
$$\boldsymbol{w}_{\boldsymbol{\lambda}} = \arg\min_{\boldsymbol{w}} \lambda_1 L_{0,0}(\boldsymbol{w}) + (\tfrac{m_{0,\star}}{m} - \lambda_1)L_{0,1}(\boldsymbol{w}) + \lambda_2 L_{1,0}(\boldsymbol{w}) + (\tfrac{m_{1,\star}}{m} - \lambda_2)L_{1,1}(\boldsymbol{w}).$$

**Demographic parity** Demographic parity (Feldman et al., 2015) is satisfied if two sensitive groups have equal positive prediction rates. If $m_{y,z}$'s are all equal, then $L_{0,0}(\boldsymbol{w}) = L_{1,0}(\boldsymbol{w})$ and $L_{0,1}(\boldsymbol{w}) = L_{1,1}(\boldsymbol{w})$ can serve as a sufficient condition for demographic parity; see Sec. A.1 for why and how to handle demographic parity when this condition does not hold. To satisfy this sufficient condition, we now adjust (i) the the sampling probability between $L_{0,0}(\boldsymbol{w})$ and $L_{1,0}(\boldsymbol{w})$ and (ii) the the sampling probability between $L_{0,1}(\boldsymbol{w})$ and $L_{1,1}(\boldsymbol{w})$. This gives us the following bilevel optimization problem:

$$\min_{\boldsymbol{\lambda} \in [0, \frac{m_{\star,0}}{m}] \times [0, \frac{m_{\star,1}}{m}]} \max\{|L_{0,0}(\boldsymbol{w}_{\boldsymbol{\lambda}}) - L_{1,0}(\boldsymbol{w}_{\boldsymbol{\lambda}})|, |L_{0,1}(\boldsymbol{w}_{\boldsymbol{\lambda}}) - L_{1,1}(\boldsymbol{w}_{\boldsymbol{\lambda}})|\},$$
$$\boldsymbol{w}_{\boldsymbol{\lambda}} = \arg\min_{\boldsymbol{w}} \lambda_1 L_{0,0}(\boldsymbol{w}) + (\tfrac{m_{\star,0}}{m} - \lambda_1)L_{1,0}(\boldsymbol{w}) + \lambda_2 L_{0,1}(\boldsymbol{w}) + (\tfrac{m_{\star,1}}{m} - \lambda_2)L_{1,1}(\boldsymbol{w}).$$

**Beyond binary labels/sensitive attributes** While the previous examples assumed binary-valued labels and sensitive attributes, our framework is applicable to the cases where the alphabet sizes are beyond binary. As an example, consider the equal opportunity criterion when $\mathbb{Z} = \{0, 1, \ldots, n_z - 1\}$. The condition reads $L_{1,0}(\boldsymbol{w}) = L_{1,1}(\boldsymbol{w}) = \cdots = L_{1,n_z-1}(\boldsymbol{w})$. To satisfy this condition, we adjust the sampling probability between $L_{1,j}(\boldsymbol{w})$'s by introducing $\binom{n_z}{2}$-dimensional outer optimization variable $\boldsymbol{\lambda}$, and design the outer objective function as $\max_{j_1,j_2 \in \mathbb{Z}} |L_{1,j_1}(\boldsymbol{w}) - L_{1,j_2}(\boldsymbol{w})|$. In our implementation, however, we only use $(n_z - 1)$-dimensional disparity objectives as an approximation (i.e., $\max_{j_1 \in \{0,1,\ldots,n_z-2\}} |L_{1,j_1}(\boldsymbol{w}) - L_{1,j_1+1}(\boldsymbol{w})|$) for better efficiency. Suppose the level of disparity is $\epsilon$ when FairBatch compares all possible combination pairs of sensitive groups. Now suppose we only optimize on the sequential $(n_z - 1)$ disparity objectives. Then we will fail to ensure that other objectives like $|L_{1,3}(\boldsymbol{w}) - L_{1,1}(\boldsymbol{w})|$ are within $\epsilon$. In the worst case, the objective $|L_{1,n_z-1}(\boldsymbol{w}) - L_{1,1}(\boldsymbol{w})|$ may be $(n_z - 1) \times \epsilon$, as we only guarantee that each $|L_{1,j_1}(\boldsymbol{w}) - L_{1,j_1+1}(\boldsymbol{w})| \leq \epsilon$. If $\epsilon$ is small enough, the disparity of our approximation becomes reasonable as well. One can also handle other fairness criteria in a similar way.

## 3 UPDATE RULE OF FAIRBATCH

We design efficient update rules of FairBatch for different numbers of disparities. Let us define $d$ as the dimension of the outer optimization variable $\boldsymbol{\lambda}$, which is the same as the total number of disparities. We first analyze the simplest case where $d = 1$. We show that a simple gradient descent algorithm can provably solve the outer optimization problem. The equal opportunity example in the previous section falls in this category. We then extend the algorithm developed for the one-dimensional case to the multi-dimensional ($d > 1$) case. Equalized odds and demographic parity fall in this category.

### 3.1 Update Rule for $d = 1$

When $d = 1$, the general form of our bilevel optimization problem can be written as follows:

$$\min_{\lambda \in [0, c_1]} |f_1(\boldsymbol{w}_\lambda) - g_1(\boldsymbol{w}_\lambda)|, \ \boldsymbol{w}_\lambda = \arg\min_{\boldsymbol{w}} \lambda f_1(\boldsymbol{w}) + (c_1 - \lambda) g_1(\boldsymbol{w}) + h(\boldsymbol{w}),$$

where $c_1 > 0$ a constant. Let $F(\lambda) = |f_1(\boldsymbol{w}_\lambda) - g_1(\boldsymbol{w}_\lambda)|$. The following lemma shows that $F(\lambda)$ is *quasiconvex* in $\lambda$ under some mild conditions, and its signed gradient can be efficiently computed.

**Lemma 1** (Quasi-convexity of $F(\lambda)$). *For $d = 1$, if $f_1(\cdot)$, $g_1(\cdot)$, and $h(\cdot)$ satisfy*

1. $h(\boldsymbol{w}) = 0$ *or*

2. *if $f_1(\cdot)$, $g_1(\cdot)$, and $h(\cdot)$ are twice differentiable, $\lambda \nabla^2 f_1(\boldsymbol{w}_\lambda) + (c_1 - \lambda) \nabla^2 g_1(\boldsymbol{w}_\lambda) + \nabla^2 h(\boldsymbol{w}_\lambda) \succ 0$ for every $\lambda \in [0, c_1]$,*

*then $F(\lambda)$ is quasi-convex, i.e., $F(t\lambda + (1 - t)\lambda') \leq \max \left\{ F(\lambda), F(\lambda') \right\}$ for all $t \in [0, 1]$ and $\lambda, \lambda'$. Also, if $F(\cdot) \neq 0$, then $\partial_\lambda F(\lambda) = \{v\}$ and $\mathrm{sign}(v) = \mathrm{sign}(g_1(\boldsymbol{w}_\lambda) - f_1(\boldsymbol{w}_\lambda))$.*

**Remark 1.** *The quasiconvexity of $F(\lambda)$ is valid when at least one of the conditions in Lemma 1 holds. For the second condition, if $f_1(\cdot)$, $g_1(\cdot)$, and $h(\cdot)$ are convex, this condition will hold unless all the three functions share their stationary points, which is very unlikely. While there is no theoretical guarantee for the non-convex settings, FairBatch still shows on par or better results than the other fairness approaches in general settings where the functions may not be convex (see Sec. 4).*

The proof for Lemma 1 can be found in Sec. A.2. Note that quasiconvexity immediately implies a unique minimum (Boyd et al., 2004). Thus, we design the following signed gradient-based optimization algorithm:

$$\forall t \in \{0, 1, \ldots\} : \lambda^{(t+1)} = \lambda^{(t)} - \alpha \cdot \mathrm{sign}(g_1(\boldsymbol{w}_\lambda) - f_1(\boldsymbol{w}_\lambda)).$$

This algorithm increases $\lambda$ by $\alpha$ if $f_1(\boldsymbol{w}_\lambda) \leq g_1(\boldsymbol{w}_\lambda)$ and decreases $\lambda$ by $\alpha$ otherwise. Recall that this is consistent with our intuition: It increases the sampling probability of a disadvantageous group and decreases that of an advantageous group. The following proposition shows that the proposed algorithm converges to the optimal solution.

**Proposition 1.** *Let $\lambda^* = \arg\min_\lambda F(\lambda)$ and $t \in \mathbb{Z}^{0+}$. Then, $|\lambda^{(t)} - \lambda^*| \leq \max\{|\lambda^{(0)} - \lambda^*| - t\alpha, \alpha\}$.*

**Remark 2.** *$F(\lambda)$ is not necessarily convex even when we assume the inner objective functions $f_1(\cdot)$ and $g_1(\cdot)$ are convex or even strongly convex. See Sec. A.3 for a counter example.*

### 3.2 Update Rule for $d \geq 1$

We now develop an efficient update algorithm for the following general bilevel optimization:

$$\min_{\boldsymbol{\lambda} \in \Lambda} \max_{i=1,\ldots,d} |f_i(\boldsymbol{w}_{\boldsymbol{\lambda}}) - g_i(\boldsymbol{w}_{\boldsymbol{\lambda}})|, \quad \boldsymbol{w}_{\boldsymbol{\lambda}} = \arg\min_{\boldsymbol{w}} \sum_{i=1}^d [\lambda_i f_i(\boldsymbol{w}) + (c_i - \lambda_i) g_i(\boldsymbol{w})] + h(\boldsymbol{w}).$$

Here, $\Lambda = [0, c_1] \times [0, c_2] \times \cdots \times [0, c_d]$, where $c_i$'s are some positive constants. Denoting by $F(\boldsymbol{\lambda})$ the outer objective function, let us first derive the gradient of it. Under some mild conditions (see Sec. A.4) on $f_i(\cdot)$'s, $g_i(\cdot)$'s, and $h(\cdot)$:

$$\gamma_i := \mathrm{sign}(g_{i^*}(\boldsymbol{w}) - f_{i^*}(\boldsymbol{w}))(\nabla f_{i^*}(\boldsymbol{w}) - \nabla g_{i^*}(\boldsymbol{w}))^\top \boldsymbol{H}_{\boldsymbol{\lambda}}^{-1}(\nabla f_i(\boldsymbol{w}) - \nabla g_i(\boldsymbol{w})) \in \partial_{\lambda_i} F(\boldsymbol{\lambda}), \ \forall i,$$

where $i^* = \arg\max_i |f_i(\boldsymbol{w}) - g_i(\boldsymbol{w})|$, and $\boldsymbol{H}_{\boldsymbol{\lambda}}$ is positive definite. See Sec. A.4 for the derivation. Since subdifferential is always a convex set, it follows that $\boldsymbol{\gamma} := (\gamma_1, \gamma_2, \ldots, \gamma_d) \in \partial_{\boldsymbol{\lambda}} F(\boldsymbol{\lambda})$. Computing the subgradient $\boldsymbol{\gamma}$ requires us to compute $\boldsymbol{H}_{\boldsymbol{\lambda}}$, which involves the Hessian matrices of the inner objective function. To avoid this expensive computation, we approximate $\boldsymbol{\gamma} \approx (0, 0, \ldots, \gamma_{i^*}, \ldots, 0)$. See Sec. A.5 for the rationale and intuition behind this approximation. Then, similar to the case of $d = 1$, we have $\mathrm{sign}(\boldsymbol{\gamma}) = (0, 0, \ldots, \mathrm{sign}(g_{i^*}(\boldsymbol{w}_\lambda) - f_{i^*}(\boldsymbol{w}_\lambda)), 0, \ldots, 0)$. This gives us the general update rule of FairBatch (see Sec. A.6 for pseudocode):

$$\forall t \in \{0, 1, \ldots\} : \lambda_{i^*}^{(t+1)} = \lambda_{i^*}^{(t)} - \alpha \cdot \mathrm{sign}(g_{i^*}(\boldsymbol{w}_\lambda) - f_{i^*}(\boldsymbol{w}_\lambda)), \ \lambda_i^{(t+1)} = \lambda_i^{(t)}, \ \forall i \neq i^*.$$

## 4 EXPERIMENTS

We use logistic regression in all experiments except for Sec. 4.2 where we fine-tune ResNet18 (He et al., 2016) and GoogLeNet (Szegedy et al., 2015) in order to demonstrate FairBatch's ability to improve fairness of pre-trained models. We evaluate all models on separate test sets and repeat all experiments with 10 different random seeds. We use PyTorch, and our experiments are performed on a server with Intel i7-6850 CPUs and NVIDIA TITAN Xp GPUs. See Sec. B.1 for more details.

**Measuring Fairness** Here we first focus on the equal opportunity (EO) and demographic parity (DP) measures in Sec. 4.1 and Sec. 4.3. The equalized odds (ED) measure is used in Sec. 4.2 and Sec. B.2. To quantify EO, ED, and DP, we compute the disparity between sensitive groups: *EO disparity* $= \max_{z \in \mathbb{Z}} |\Pr(\hat{y} = 1 | z = z, y = 1) - \Pr(\hat{y} = 1 | y = 1)|$, *ED disparity* $= \max_{z \in \mathbb{Z}, y \in \mathbb{Y}, \hat{y} \in \hat{\mathbb{Y}}} |\Pr(\hat{y} = \hat{y} | z = z, y = y) - \Pr(\hat{y} = \hat{y} | y = y)|$, and *DP disparity* $= \max_{z \in \mathbb{Z}} |\Pr(\hat{y} = 1 | z = z) - \Pr(\hat{y} = 1)|$. As we discussed in Sec. 3, EO has a single-dimension outer optimization where the number of disparities $d = 1$ while ED and DP have multi-dimensional outer optimizations where $d > 1$.

**Datasets** We generate a synthetic dataset of 3,000 examples with two non-sensitive attributes ($x_1$, $x_2$), a binary sensitive attribute z, and a binary label y, using a method similar to the one in (Zafar et al., 2017a). A tuple ($x_1$, $x_2$, y) is randomly generated based on the two Gaussian distributions: $(x_1, x_2)|y = 0 \sim \mathcal{N}([-2; -2], [10, 1; 1, 3])$ and $(x_1, x_2)|y = 1 \sim \mathcal{N}([2; 2], [5, 1; 1, 5])$. For z, we generate biased data using an *unfair scenario* $\Pr(z = 1) = \Pr((x_1', x_2')|y = 1)/[\Pr((x_1', x_2')|y = 0) + \Pr((x_1', x_2')|y = 1)]$ where $(x_1', x_2') = (x_1 \cos(\pi/4) - x_2 \sin(\pi/4), x_1 \sin(\pi/4) + x_2 \cos(\pi/4))$.

We use the real benchmark datasets: ProPublica COMPAS (Angwin et al., 2016) and AdultCensus (Kohavi, 1996) datasets with 5,278 and 43,131 examples, respectively. We use the same pre-processing as in IBM's AI Fairness 360 (Bellamy et al., 2019) and use GENDER as the sensitive attribute. We also employ the UTKFace dataset (Zhang et al., 2017) with 23,708 images to demonstrate the fine-tuning ability of FairBatch in Sec. 4.2.

**Baselines** We employ three types of baselines: (1) non-fair training with logistic regression (LR); (2) fair training via pre-processing; and (3) fair training via in-processing.

For pre-processing methods, we first consider a simple approach that we call Cutting, which evens the data sizes of sensitive groups via saturating them to the smallest-group data size. One can think of a similar alternative approach: *Boosting* all of the smaller-group data sizes to the largest one, but we do not report herein due to similar performances that we found relative to Cutting. The other two are the state of the arts: reweighing (Kamiran and Calders, 2011) (RW) and Label Bias Correction (Jiang and Nachum, 2020) (LBC). RW intends to balance importance levels across sensitive groups via example weighting, but sticks with these weights throughout the entire model training, unlike FairBatch. LBC iteratively trains an entire model with example weighting towards an unbiased data distribution.

For in-processing methods, we compare with the following three: Fairness Constraints (Zafar et al., 2017a;b) (FC), Adversarial Debiasing (Zhang et al., 2018) (AD), and AdaFair (Iosifidis and Ntoutsi, 2019). FC incorporates a regularization term in an effort to reduce the disparities among sensitive groups. AD is an adversarial learning approach that intends to maximize the independence between the predicted labels and sensitive attributes. In our experiments, a slight modification is made to AD for improving training stability: Not employing one regularization term used for restricting the training direction. AdaFair is an ensemble technique that equips the prominent AdaBoost (Friedman et al., 2000) with a fairness aspect. Here the examples that lead to unfair and inaccurate performances are considered to be the *difficult* instances. In our experiments, natural generalization of AdaFair intended for ED is made to encompass EO and DP; see Sec. B.3 for the generalization. While AdaFair bears spiritual similarity to FairBatch in a sense that mistreated examples are weighted progressively, it comes with a significant distinction in update scale. It is basically a boosting technique; hence such updates are done in distinctive predictors through different *rounds*; see Sec. 5 for details.

**FairBatch Settings** To set $\alpha$, we start from a candidate set of values within the range [0.0001, 0.05] and use cross-validation on the training set to choose the value that results in the highest accuracy with low fairness violation. The default batch sizes are: 100 (synthetic); 200 (COMPAS), 1,000 (AdultCensus); and 32 (UTKFace).

Table 1: Performances on the synthetic, COMPAS, and AdultCensus test sets w.r.t. equal opportunity (EO). We compare FairBatch with three types of baselines: (1) non-fair method: LR; (2) fair training via pre-processing: Cutting, RW (Kamiran and Calders, 2011), and LBC (Jiang and Nachum, 2020); (3) fair training via in-processing: FC (Zafar et al., 2017b), AD (Zhang et al., 2018), and AdaFair (Iosifidis and Ntoutsi, 2019). Experiments are repeated 10 times.

| | Synthetic | | | COMPAS | | | AdultCensus | | |
|---|---|---|---|---|---|---|---|---|---|
| Method | Acc. | EO Disp. | Epochs | Acc. | EO Disp. | Epochs | Acc. | EO Disp. | Epochs |
| LR | .885±.000 | .115±.000 | 400 | .681±.002 | .239±.006 | 300 | .845±.001 | .054±.005 | 300 |
| Cutting | .858±.001 | .028±.002 | 800 | .674±.005 | .055±.018 | 600 | .802±.002 | .054±.007 | 600 |
| RW | .858±.000 | .020±.000 | 800 | .685±.000 | .137±.000 | 300 | .835±.001 | .134±.006 | 100 |
| LBC | .858±.001 | .022±.000 | 11200 | .673±.002 | .031±.006 | 3900 | .841±.003 | **.011±.003** | 6300 |
| FC | .833±.001 | **.007±.000** | 700 | .656±.006 | .059±.028 | 100 | .844±.001 | .021±.004 | 300 |
| AD | .837±.010 | .026±.007 | 200 | .683±.001 | .067±.029 | 300 | .841±.003 | .016±.005 | 400 |
| AdaFair | .868±.000 | .043±.001 | 16000 | .664±.004 | **.018±.004** | 9600 | .844±.001 | .038±.004 | 9000 |
| **FairBatch** | .855±.000 | .012±.001 | 300 | .681±.001 | .022±.005 | 100 | .844±.001 | **.011±.003** | 400 |

Table 2: Performances on the synthetic, COMPAS, and AdultCensus test sets w.r.t. demographic parity (DP). The other settings are identical to those in Table 1.

| | Synthetic | | | COMPAS | | | AdultCensus | | |
|---|---|---|---|---|---|---|---|---|---|
| Method | Acc. | DP Disp. | Epochs | Acc. | DP Disp. | Epochs | Acc. | DP Disp. | Epochs |
| LR | .885±.000 | .257±.000 | 400 | .681±.002 | .192±.006 | 300 | .845±.001 | .125±.001 | 300 |
| Cutting | .885±.001 | .258±.001 | 500 | .677±.004 | .205±.025 | 400 | .846±.001 | .123±.002 | 300 |
| RW | .857±.000 | .164±.001 | 400 | .685±.000 | .103±.000 | 300 | .835±.001 | .052±.003 | 300 |
| LBC | .768±.000 | .042±.001 | 16000 | .671±.002 | **.032±.009** | 7800 | .815±.003 | .011±.002 | 12600 |
| FC | .785±.013 | .058±.010 | 600 | .684±.001 | .083±.015 | 70 | .812±.009 | .025±.006 | 100 |
| AD | .812±.008 | .063±.014 | 700 | .683±.002 | .054±.019 | 550 | .815±.008 | .018±.004 | 400 |
| AdaFair | .784±.001 | .089±.001 | 52000 | .642±.004 | .033±.011 | 6300 | .825±.002 | .040±.001 | 27000 |
| **FairBatch** | .794±.001 | **.040±.001** | 450 | .681±.001 | .036±.023 | 300 | .823±.001 | **.010±.005** | 600 |

## 4.1 Accuracy, Fairness, and Runtime

Table 1 compares FairBatch against the other approaches on the synthetic, COMPAS, and AdultCensus test sets w.r.t. accuracy, EO disparity, and complexity (reflected in the number of epochs). In Sec. B.4, we also present the convergence plot of EO disparity as a function of the number of epochs. LR in row 1 is logistic regression without any fairness technique. The pre-processing techniques in rows 2–4 reduce EO disparity yet while sacrificing the accuracy performance. The in-processing techniques in rows 5–7 further reduce EO disparity yet still sacrificing accuracy. FairBatch, presented in the last row, offers comparable (or even greater) fairness performance while sacrificing less accuracy. We also present accuracy and fairness trade-off curves of FairBatch in Sec. B.5. One key implementation benefit is reflected in the small numbers of epochs. We also obtain consistent wall clock times, presented in Sec. B.6. As mentioned earlier, AdaFair is the most similar in spirit to FairBatch as it adjusts example weights based on the fairness performances of prior models. We demonstrate in Sec. B.7 that FairBatch and AdaFair indeed show similar convergence behaviors yet in different scales (rounds for AdaFair vs. epochs for FairBatch). One distinctive feature of FairBatch relative to AdaFair is the use of a *single* model training, thus enabling much faster speed (22.5–96x). We also make similar comparisons yet w.r.t. another fairness measure: DP disparity. See Table 2. Recall that minimizing DP disparity involves adjusting two hyperparameters ($\lambda_1, \lambda_2$), which also means that $d = 2$. Although FairBatch's theoretical guarantees hold only when using one hyperparameter (i.e., $d = 1$), we nonetheless see similar results where FairBatch is on par or better than the other approaches, while being the most robust in all aspects.

Table 3: Performances of the pre-trained models fine-tuned with FairBatch on the UTKFace test set w.r.t. equalized odds (ED) for two fairness scenarios. While Tables 1 and 2 already demonstrate FairBatch's performance against the state of the arts, the emphasis here is more on FairBatch's usability where it is easy to adopt and yet improves the fairness of existing models.

| Pre-trained model | Method | z: RACE, y: GENDER | | | z: RACE, y: AGE | | |
|---|---|---|---|---|---|---|---|
| | | Acc. | ED Disp. | Epochs | Acc. | ED Disp. | Epochs |
| ResNet18 | Original | .893±.002 | .086±.012 | 19 | .722±.011 | .311±.053 | 10 |
| | Cutting | .592±.020 | .099±.014 | 18 | .466±.018 | **.139±.021** | 20 |
| | **FairBatch** | .894±.002 | **.063±.013** | 30 | .758±.004 | .220±.016 | 10 |
| GoogLeNet | Original | .888±.003 | .105±.016 | 20 | .746±.006 | .294±.034 | 14 |
| | Cutting | .606±.010 | .076±.017 | 20 | .495±.017 | **.168±.033** | 9 |
| | **FairBatch** | .891±.002 | **.061±.006** | 11 | .741±.018 | .202±.019 | 8 |

## 4.2 FINE-TUNING PRETRAINED UNFAIR MODELS FOR FAIRNESS

While Tables 1 and 2 already demonstrate FairBatch's performance against the state of the arts, in this section we emphasize the *usability* of FairBatch by showing how it can improve fairness of any pretrained unfair model via fine-tuning and only compare it with Cutting, which is also easy to adopt. Table 3 shows how FairBatch improves fairness of pre-trained models (ResNet18 (He et al., 2016) and GoogLeNet (Szegedy et al., 2015)) on the UTKFace dataset (Zhang et al., 2017). Each image has three types of attributes: GENDER, RACE, and AGE. We use RACE as the sensitive attribute and consider two scenarios where the label attribute is GENDER or AGE. While GENDER is binary, AGE is multi-valued ($<$21, 21–40, 41–60, and $>$60), so we extend FairBatch in a straightforward fashion; see Sec. B.8 for details. Both Cutting and FairBatch reduce the ED disparities of the original pre-trained models. However, only FairBatch does so without sacrificing accuracy performance.

## 4.3 COMPATIBILITY WITH OTHER BATCH SELECTION TECHNIQUES

We demonstrate another key aspect of FairBatch: *Compatibility* with existing batch selection approaches that use importance sampling for faster convergence in training. The key functionality of the prior batch selection techniques is that examples considered to be "important" are given higher weights so as to be sampled more frequently. FairBatch can easily be tuned to accommodate such functionality: determining the batch-ratios of sensitive groups and then sampling using the importance weights per group. We evaluate FairBatch combined with one prominent technique, loss-based weighting (Loshchilov and Hutter, 2016), on our synthetic dataset using EO and DP. We find that FairBatch indeed converges more quickly. It uses about 50 fewer epochs with similar fairness performances; see Sec. B.9 for the EO and DP convergence plots.

## 5 RELATED WORK

**Model Fairness** Various fairness measures have been proposed to reflect legal and social issues (Narayanan, 2018). Among them, we focus on group fairness measures: equal opportunity (Hardt et al., 2016), equalized odds (Hardt et al., 2016), and demographic parity (Feldman et al., 2015). A variety of techniques have been proposed and can be categorized into (1) pre-processing techniques (Kamiran and Calders, 2011; Zemel et al., 2013; Feldman et al., 2015; du Pin Calmon et al., 2017; Choi et al., 2020; Jiang and Nachum, 2020), which debias or reweight data, (2) in-processing techniques (Kamishima et al., 2012; Zafar et al., 2017a;b; Agarwal et al., 2018; Zhang et al., 2018; Cotter et al., 2019; Roh et al., 2020), which tailor the model training for fairness, and (3) post-processing techniques (Kamiran et al., 2012; Hardt et al., 2016; Pleiss et al., 2017; Chzhen et al., 2019), which perturb only the model output without touching upon the inside. Most of these methods require broad changes in data preprocessing, model training, or model outputs in machine learning systems (Venkatasubramanian, 2019). In contrast, FairBatch only requires a single-line change in code to replace batch selection while achieving comparable or even greater performances against the state of the arts.

Among the fairness techniques, AdaFair (Iosifidis and Ntoutsi, 2019) is the most similar in spirit to FairBatch. AdaFair extends the well-known AdaBoost (Friedman et al., 2000) where examples that lead to poor accuracy or fairness are boosted, i.e., given higher weights during the next round of training a new model that is added to the ensemble. In comparison, FairBatch is based on theoretical foundations of bilevel optimization and effectively performs the reweighting *during each epoch* (not through rounds), which leads to an order of magnitude improvement in speed as shown in Sec. 4.1.

Although not our immediate focus, there are other noteworthy fairness measures: (1) individual fairness (Dwork et al., 2012) where close examples should be treated similarly, (2) causality-based fairness (Kilbertus et al., 2017; Kusner et al., 2017; Zhang and Bareinboim, 2018; Nabi and Shpitser, 2018; Khademi et al., 2019), which aims to overcome the limitations of non-causal approaches by understanding the causal relationship between attributes, and (3) distributionally robust optimization (DRO) (Sinha et al., 2017)-based fairness (Hashimoto et al., 2018), which achieves accuracy parity without the knowledge of sensitive attribute by balancing the risks across all distributions. Extending FairBatch to support these measures is an interesting future work.

Finally, Chouldechova and Roth (2018) describe three causes of unfairness that help clarify FairBatch's fairness contributions: (1) minimizing average error fits majority populations, (2) bias encoded in data, and (3) the need to explore and gather more data. FairBatch addresses the cause (1) via balancing the sensitive group ratios within a batch. FairBatch also addresses (2) in some cases. For example, consider the recidivism prediction problem described in (Chouldechova and Roth, 2018) where minority populations have biased labels. In this case, FairBatch can be configured to make the recidivism prediction rate for the minority population similar to those of other populations. There may be other types of data bias that FairBatch is not able to address. Finally, FairBatch does not directly address (3) where one must gather more data for better fairness. Instead, there is a recent line of work that studies data collection techniques (Tae and Whang, 2021) for fairness.

**Batch Selection**  The batch selection literature for SGD focuses on analyzing the effect of batch sizes (Keskar et al., 2017; Masters and Luschi, 2018) and various sampling techniques (Shamir, 2016; Gürbüzbalaban et al., 2019). More recently, importance sampling techniques have been proposed for faster convergence (Loshchilov and Hutter, 2016; Alain et al., 2016; Stich et al., 2017; Csiba and Richtárik, 2018; Katharopoulos and Fleuret, 2018; Johnson and Guestrin, 2018). In comparison, FairBatch takes the novel approach of using batch selection for better fairness and is compatible with other existing techniques.

## 6  CONCLUSION

We addressed model fairness via the lens of bilevel optimization and proposed the FairBatch batch selection algorithm. The bilevel optimization provides a natural framework where the inner optimizer is SGD, and the outer optimizer performs adaptive batch selection to improve fairness. We presented FairBatch for implementing this optimization and showed how its underlying theory supports the fairness measures: equal opportunity, equalized odds, and demographic parity. We showed that FairBatch offers respectful performances that are on par or even better than the state of the arts w.r.t. all aspects in consideration: accuracy, fairness, and runtime. Also, FairBatch can readily be adopted to machine learning systems with a minimal change of replacing the batch selection with a single-line of code and be gracefully merged with other batch selection techniques used for faster convergence.

## ACKNOWLEDGEMENTS

Yuji Roh and Steven E. Whang were supported by a Google AI Focused Research Award and by the Engineering Research Center Program through the National Research Foundation of Korea (NRF) funded by the Korean Government MSIT (NRF-2018R1A5A1059921). Kangwook Lee was supported by NSF/Intel Partnership on Machine Learning for Wireless Networking Program under Grant No. CNS-2003129. Changho Suh was supported by Institute for Information & communications Technology Planning & Evaluation (IITP) grant funded by the Korea government (MSIT) (No. 2019-0-01396, Development of framework for analyzing, detecting, mitigating of bias in AI model and training data).

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

## A  APPENDIX – THEORY AND ALGORITHMS

### A.1  DEMOGRAPHIC PARITY

We continue from Sec. 2 and provide more details on how we can capture demographic parity using our bilevel optimization framework.

**Proposition 2.** *If $m_{0,0} = m_{0,1} = m_{1,0} = m_{1,1}$, then $L_{0,0}(\boldsymbol{w}) = L_{1,0}(\boldsymbol{w})$ and $L_{0,1}(\boldsymbol{w}) = L_{1,1}(\boldsymbol{w})$ can serve as a sufficient condition for demographic parity.*

*Proof.* Slightly abusing the notation, we denote by $\Pr(\cdot)$ the empirical probability. The demographic parity is satisfied when $\Pr(\hat{y} = 1 | z = 0) = \Pr(\hat{y} = 1 | z = 1)$ holds. Thus,

$$\Pr(\hat{y} = 1, y = 0 | z = 0) + \Pr(\hat{y} = 1, y = 1 | z = 0) = \Pr(\hat{y} = 1, y = 0 | z = 1) + \Pr(\hat{y} = 1, y = 1 | z = 1).$$

Since $\ell(|1 - y|, \cdot) = 1 - \ell(y, \cdot)$, we have

$$\frac{1}{m_{\star,0}} \sum_{i:y_i=0,z_i=0} (1 - \ell(y_i, \hat{y}_i)) + \frac{1}{m_{\star,0}} \sum_{i:y_i=1,z_i=0} \ell(y_i, \hat{y}_i)$$
$$= \frac{1}{m_{\star,1}} \sum_{i:y_i=0,z_i=1} (1 - \ell(y_i, \hat{y}_i)) + \frac{1}{m_{\star,1}} \sum_{i:y_i=1,z_i=1} \ell(y_i, \hat{y}_i).$$

By replacing $\sum_{i:y_i=y,z_i=z} \ell(y_i, \hat{y}_i) = m_{y,z} L_{y,z}(\boldsymbol{w})$,

$$\frac{m_{0,0}}{m_{\star,0}}(1 - L_{0,0}(\boldsymbol{w})) + \frac{m_{1,0}}{m_{\star,0}} L_{1,0}(\boldsymbol{w}) = \frac{m_{0,1}}{m_{\star,1}}(1 - L_{0,1}(\boldsymbol{w})) + \frac{m_{1,1}}{m_{\star,1}} L_{1,1}(\boldsymbol{w}).$$

If $m_{0,0} = m_{0,1} = m_{1,0} = m_{1,1}$, this reduces to $L_{0,0}(\boldsymbol{w}) = L_{1,0}(\boldsymbol{w})$ and $L_{0,1}(\boldsymbol{w}) = L_{1,1}(\boldsymbol{w})$, the above condition reduces to

$$-L_{0,0}(\boldsymbol{w}) + L_{1,0}(\boldsymbol{w}) = -L_{0,1}(\boldsymbol{w}) + L_{1,1}(\boldsymbol{w}).$$

A sufficient condition to the above condition is $L_{0,0}(\boldsymbol{w}) = L_{1,0}(\boldsymbol{w})$ and $L_{0,1}(\boldsymbol{w}) = L_{1,1}(\boldsymbol{w})$. □

In general, the condition of the above proposition does not hold. Observe that another sufficient condition to demographic parity is as follows:

$$\frac{m_{1,0}}{m_{\star,0}} L_{1,0}(\boldsymbol{w}) - \frac{m_{1,1}}{m_{\star,1}} L_{1,1}(\boldsymbol{w}) = 0$$
$$\frac{m_{0,0}}{m_{\star,0}} L_{0,0}(\boldsymbol{w}) - \frac{m_{0,1}}{m_{\star,1}} L_{0,1}(\boldsymbol{w}) = \frac{m_{0,0}}{m_{\star,0}} - \frac{m_{0,1}}{m_{\star,1}}$$

Let us define $L'_{1,0}(\boldsymbol{w}) = \frac{m_{1,0}}{m_{\star,0}} L_{1,0}(\boldsymbol{w})$, $L'_{1,1}(\boldsymbol{w}) = \frac{m_{1,1}}{m_{\star,1}} L_{1,1}(\boldsymbol{w})$, $L'_{0,0}(\boldsymbol{w}) = \frac{m_{0,0}}{m_{\star,0}} L_{0,0}(\boldsymbol{w})$, $L'_{0,1}(\boldsymbol{w}) = \frac{m_{0,1}}{m_{\star,1}} L_{0,1}(\boldsymbol{w})$, and $c = \frac{m_{0,0}}{m_{\star,0}} - \frac{m_{0,1}}{m_{\star,1}}$. Also, define $|x|_c = \max\{x - c, c - x\}$. Then, we have the following bilevel optimization problem:

$$\min_{\boldsymbol{\lambda} \in [0,1] \times [0,1]} \max\{|L'_{1,0}(\boldsymbol{w}_{\boldsymbol{\lambda}}) - L'_{1,1}(\boldsymbol{w}_{\boldsymbol{\lambda}})|, |L'_{0,0}(\boldsymbol{w}_{\boldsymbol{\lambda}}) - L'_{0,1}(\boldsymbol{w}_{\boldsymbol{\lambda}})|_c\},$$
$$\boldsymbol{w}_{\boldsymbol{\lambda}} = \arg\min_{\boldsymbol{w}} \lambda_1 L'_{0,0}(\boldsymbol{w}) + (1 - \lambda_1) L'_{1,0}(\boldsymbol{w}) + \lambda_2 L'_{0,1}(\boldsymbol{w}) + (1 - \lambda_2) L'_{1,1}(\boldsymbol{w}).$$

### A.2  PROOF FOR LEMMA 1

We continue from Sec. 3.1 and provide a full proof for Lemma 1. Here we recall Lemma 1.

**Lemma 1** (Quasi-convexity of $F(\lambda)$)**.** *For $d = 1$, if $f_1(\cdot)$, $g_1(\cdot)$, and $h(\cdot)$ satisfy*

1. *$h(\boldsymbol{w}) = 0$  or*

2. *if $f_1(\cdot)$, $g_1(\cdot)$, and $h(\cdot)$ are twice differentiable, $\lambda \nabla^2 f_1(\boldsymbol{w}_\lambda) + (c_1 - \lambda) \nabla^2 g_1(\boldsymbol{w}_\lambda) + \nabla^2 h(\boldsymbol{w}_\lambda) \succ 0$ for every $\lambda \in [0, c_1]$,*

*then $F(\lambda)$ is quasi-convex, i.e., $F(t\lambda + (1 - t)\lambda') \leq \max\{F(\lambda), F(\lambda')\}$ for all $t \in [0, 1]$ and $\lambda, \lambda'$. Also, if $F(\cdot) \neq 0$, then $\partial_\lambda F(\lambda) = \{v\}$ and $\text{sign}(v) = \text{sign}(g_1(\boldsymbol{w}_\lambda) - f_1(\boldsymbol{w}_\lambda))$.*

*Proof.* It it known that a continuous function $f : \mathbb{R} \to \mathbb{R}$ is quasiconvex if and only if at least one of the following conditions holds: 1) nondecreasing, 2) nonincreasing, and 3) nonincreasing and then nondecreasing (Boyd et al., 2004). We will prove the lemma by showing that the function $F(\lambda)$ is quasiconvex by showing that it is nonincreasing and then nondecreasing. More precisely, we will show that $f_1(\boldsymbol{w}_\lambda) - g_1(\boldsymbol{w}_\lambda)$ is a nonincreasing function. It is easy to see that this directly implies that $|f_1(\boldsymbol{w}_\lambda) - g_1(\boldsymbol{w}_\lambda)|$ is nonincreasing and then nondecreasing.

**Case 1** ($h(\boldsymbol{w}) = 0$)    Consider $\lambda_1$ and $\lambda_2$ such that $\lambda_1 > \lambda_2$. If we can show $f_1(\boldsymbol{w}_{\lambda_1}^*) \le f_1(\boldsymbol{w}_{\lambda_2}^*)$ and $g_1(\boldsymbol{w}_{\lambda_1}^*) \ge g_1(\boldsymbol{w}_{\lambda_2}^*)$, then this implies that $f_1(\boldsymbol{w}_\lambda) - g_1(\boldsymbol{w}_\lambda)$ is a nonincreasing function. Indeed, this is very intuitive: If we increase $\lambda$, the inner optimization problems puts a higher weight on $f_1(\cdot)$, resulting in a lower value of $f_1(\boldsymbol{w}^*)$ and a higher value of $g_1(\boldsymbol{w}^*)$. We formally show this by contradiction. By the definition of $\boldsymbol{w}_\lambda$, we have the following two conditions:

$$\lambda_1 f_1(\boldsymbol{w}_{\lambda_1}^*) + (c_1 - \lambda_1) g_1(\boldsymbol{w}_{\lambda_1}^*) \le \lambda_1 f_1(\boldsymbol{w}) + (c_1 - \lambda_1) g_1(\boldsymbol{w}), \; \forall \boldsymbol{w}, \tag{1}$$

$$\lambda_2 f_1(\boldsymbol{w}_{\lambda_2}^*) + (c_1 - \lambda_2) g_1(\boldsymbol{w}_{\lambda_2}^*) \le \lambda_2 f_1(\boldsymbol{w}) + (c_1 - \lambda_2) g_1(\boldsymbol{w}), \; \forall \boldsymbol{w}. \tag{2}$$

If the lemma's statement is false, one of the following three events should occur:

1. $f_1(\boldsymbol{w}_{\lambda_1}^*) > f_1(\boldsymbol{w}_{\lambda_2}^*)$ and $g_1(\boldsymbol{w}_{\lambda_1}^*) \ge g_1(\boldsymbol{w}_{\lambda_2}^*)$: By adding these two inequalities with respective weights $\lambda_1$ and $c_1 - \lambda_1$, we have $\lambda_1 f_1(\boldsymbol{w}_{\lambda_1}^*) + (c_1 - \lambda_1) g_1(\boldsymbol{w}_{\lambda_1}^*) > \lambda_1 f_1(\boldsymbol{w}_{\lambda_2}^*) + (c_1 - \lambda_1) g_1(\boldsymbol{w}_{\lambda_2}^*)$. This contradicts equation 1.

2. $f_1(\boldsymbol{w}_{\lambda_1}^*) \le f_1(\boldsymbol{w}_{\lambda_2}^*)$ and $g_1(\boldsymbol{w}_{\lambda_1}^*) < g_1(\boldsymbol{w}_{\lambda_2}^*)$: Similarly, by adding these two inequalities with respective weights $\lambda_2$ and $c_1 - \lambda_2$, we have $\lambda_2 f_1(\boldsymbol{w}_{\lambda_2}^*) + (c_1 - \lambda_2) g_1(\boldsymbol{w}_{\lambda_2}^*) > \lambda_2 f_1(\boldsymbol{w}_{\lambda_1}^*) + (c_1 - \lambda_2) g_1(\boldsymbol{w}_{\lambda_1}^*)$. This contradicts equation 2.

3. $f_1(\boldsymbol{w}_{\lambda_1}^*) > f_1(\boldsymbol{w}_{\lambda_2}^*)$ and $g_1(\boldsymbol{w}_{\lambda_1}^*) < g_1(\boldsymbol{w}_{\lambda_2}^*)$: By adding equation 1 with $\boldsymbol{w} = \boldsymbol{w}_{\lambda_2}^*$ and equation 2 with $\boldsymbol{w} = \boldsymbol{w}_{\lambda_1}^*$, we have

$$\lambda_1 f_1(\boldsymbol{w}_{\lambda_1}^*) + (c_1 - \lambda_1) g_1(\boldsymbol{w}_{\lambda_1}^*) + \lambda_2 f_1(\boldsymbol{w}_{\lambda_2}^*) + (c_1 - \lambda_2) g_1(\boldsymbol{w}_{\lambda_2}^*)$$
$$\le \lambda_1 f_1(\boldsymbol{w}_{\lambda_2}^*) + (c_1 - \lambda_1) g_1(\boldsymbol{w}_{\lambda_2}^*) + \lambda_2 f_1(\boldsymbol{w}_{\lambda_1}^*) + (c_1 - \lambda_2) g_1(\boldsymbol{w}_{\lambda_1}^*).$$

   By rearranging terms, we have

$$(\lambda_1 - \lambda_2)(f_1(\boldsymbol{w}_{\lambda_1}^*) - f_1(\boldsymbol{w}_{\lambda_2}^*)) \le (\lambda_1 - \lambda_2)(g_1(\boldsymbol{w}_{\lambda_1}^*) - g_1(\boldsymbol{w}_{\lambda_2}^*)).$$

   By dividing both sides by $\lambda_1 - \lambda_2 > 0$, we have $f_1(\boldsymbol{w}_{\lambda_1}^*) - f_1(\boldsymbol{w}_{\lambda_2}^*) \le g_1(\boldsymbol{w}_{\lambda_1}^*) - g_1(\boldsymbol{w}_{\lambda_2}^*)$. This contradicts the condition as the left-hand side is positive while the right-hand side is negative.

This completes the proof of the first claim by contradiction.

The second claim immediately follows the first claim. Since $F(\lambda) = |f_1(\boldsymbol{w}_\lambda) - g_1(\boldsymbol{w}_\lambda)|$, we have $\frac{\mathrm{d}F(\lambda)}{\mathrm{d}\lambda} = \mathrm{sign}\,(f_1(\boldsymbol{w}_\lambda) - g_1(\boldsymbol{w}_\lambda)) \frac{\mathrm{d}}{\mathrm{d}\lambda}(f_1(\boldsymbol{w}_\lambda) - g_1(\boldsymbol{w}_\lambda))$. As shown in the earlier part of this proof, $f_1(\boldsymbol{w}_\lambda) - g_1(\boldsymbol{w}_\lambda)$ is a nonincreasing function, i.e., $\frac{\mathrm{d}f_1(\boldsymbol{w}_\lambda) - g_1(\boldsymbol{w}_\lambda)}{\mathrm{d}\lambda} \le 0$. Thus, $\mathrm{sign}(\frac{\mathrm{d}F(\lambda)}{\mathrm{d}\lambda}) = \mathrm{sign}(g_1(\boldsymbol{w}_\lambda) - f_1(\boldsymbol{w}_\lambda))$.

**Case 2** (**If** $f_1(\cdot)$, $g_1(\cdot)$, **and** $h(\cdot)$ **are twice differentiable,** $\lambda \nabla^2 f_1(\boldsymbol{w}_\lambda) + (c_1 - \lambda) \nabla^2 g_1(\boldsymbol{w}_\lambda) + \nabla^2 h(\boldsymbol{w}_\lambda) \succ 0$ **for every** $\lambda \in [0, c_1]$)    In this part of the proof, we will denote $\boldsymbol{w}_\lambda$ by $\boldsymbol{w}$ for simplicity. To show that $f_1(\boldsymbol{w}) - g_1(\boldsymbol{w})$ is a nondecreasing function (in $\lambda$), consider the derivative:

$$\frac{\mathrm{d}}{\mathrm{d}\lambda}(f_1(\boldsymbol{w}) - g_1(\boldsymbol{w})) = (\nabla f_1(\boldsymbol{w}) - \nabla g_1(\boldsymbol{w}))^\top \frac{\mathrm{d}\boldsymbol{w}}{\mathrm{d}\lambda} \tag{3}$$

To compute $\frac{\mathrm{d}\boldsymbol{w}}{\mathrm{d}\lambda}$, we implicitly differentiate (with respect to $\lambda$) the following stationary equation.

$$\lambda \nabla f_1(\boldsymbol{w}) + (c_1 - \lambda) \nabla g_1(\boldsymbol{w}) + \nabla h(\boldsymbol{w}) = 0$$
$$\Rightarrow \; \nabla f_1(\boldsymbol{w}) + \lambda \nabla^2 f_1(\boldsymbol{w}) \cdot \frac{\mathrm{d}\boldsymbol{w}}{\mathrm{d}\lambda} - \nabla g_1(\boldsymbol{w}) + (c_1 - \lambda) \nabla^2 g_1(\boldsymbol{w}) \cdot \frac{\mathrm{d}\boldsymbol{w}}{\mathrm{d}\lambda} + \nabla^2 h(\boldsymbol{w}) \cdot \frac{\mathrm{d}\boldsymbol{w}}{\mathrm{d}\lambda} = 0$$

By rearranging terms, we have

$$\left(\lambda \nabla^2 f_1(\boldsymbol{w}) + (c_1 - \lambda)\nabla^2 g_1(\boldsymbol{w}) + \nabla^2 h(\boldsymbol{w})\right)\frac{\mathrm{d}\boldsymbol{w}}{\mathrm{d}\lambda} = -(\nabla f_1(\boldsymbol{w}) - \nabla g_1(\boldsymbol{w})).$$

By the assumption, $\lambda \nabla^2 f_1(\boldsymbol{w}) + (c_1 - \lambda)\nabla^2 g_1(\boldsymbol{w}) + \nabla^2 h(\boldsymbol{w})$ is positive definite and hence invertible. Thus,

$$\frac{\mathrm{d}\boldsymbol{w}}{\mathrm{d}\lambda} = -\left(\lambda \nabla^2 f_1(\boldsymbol{w}) + (c_1 - \lambda)\nabla^2 g_1(\boldsymbol{w}) + \nabla^2 h(\boldsymbol{w})\right)^{-1}(\nabla f_1(\boldsymbol{w}) - \nabla g_1(\boldsymbol{w})).$$

Therefore,

$$\frac{\mathrm{d}}{\mathrm{d}\lambda}(f_1(\boldsymbol{w}) - g_1(\boldsymbol{w})) = -(\nabla f_1(\boldsymbol{w}) - \nabla g_1(\boldsymbol{w}))^\top \left(\lambda \nabla^2 f_1(\boldsymbol{w}) + (c_1 - \lambda)\nabla^2 g_1(\boldsymbol{w}) + \nabla^2 h(\boldsymbol{w})\right)^{-1}(\nabla f_1(\boldsymbol{w}) - \nabla g_1(\boldsymbol{w})).$$

Note that $\left(\lambda \nabla^2 f_1(\boldsymbol{w}) + (c_1 - \lambda)\nabla^2 g_1(\boldsymbol{w}) + \nabla^2 h(\boldsymbol{w})\right)^{-1}$ is also positive definite. Thus, $\frac{\mathrm{d}}{\mathrm{d}\lambda}(f_1(\boldsymbol{w}) - g_1(\boldsymbol{w}))$ is always negative, and hence $f_1(\boldsymbol{w}) - g_1(\boldsymbol{w})$ is a decreasing function.

Now, observe that

$$(f_1(\boldsymbol{w}) - g_1(\boldsymbol{w})) \cdot \frac{\mathrm{d}}{\mathrm{d}\lambda}(f_1(\boldsymbol{w}) - g_1(\boldsymbol{w})) \in \partial_\lambda F(\lambda).$$

Therefore, if $F(\cdot) \neq 0$, then $\partial_\lambda F(\lambda) = \{v\}$ and $\mathrm{sign}\,(v) = \mathrm{sign}\,(g_1(\boldsymbol{w}) - f_1(\boldsymbol{w}))$. $\qquad\square$

### A.3 INNER OBJECTIVE'S CONVEXITY DOES NOT IMPLY OUTER OBJECTIVE'S CONVEXITY

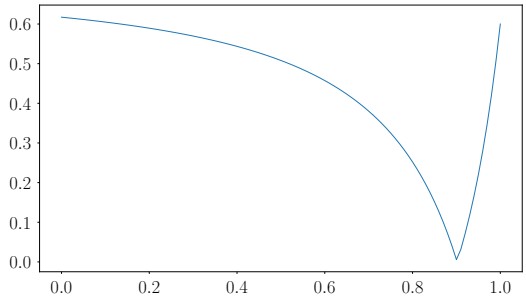

Figure 2: $F(\lambda)$ is not convex, but quasi-convex.

We continue from Sec. 3.2 and provide an example where inner objective's convexity does not imply outer objective's convexity. Consider the following strongly convex functions $f_1(\cdot)$ and $g_1(\cdot)$:

$$f_1(w) = \frac{e^w + e^{-w}}{5}, \quad g_1(w) = (w - 1)^2$$

Shown in Fig. 2 is the outer objective function $F(\lambda)$. One can observe that it is not convex. Note that it is quasiconvex by Lemma 1.

### A.4 GRADIENT WHEN $d \geq 1$

We continue from Sec. 3.2 and derive the gradient of the outer objective function. Recall how we formulated the general bilevel optimization problem:

$$\min_{\boldsymbol{\lambda} \in \Lambda} \max_{i=1,\dots,d} |f_i(\boldsymbol{w_\lambda}) - g_i(\boldsymbol{w_\lambda})|, \quad \boldsymbol{w_\lambda} = \arg\min_{\boldsymbol{w}} \sum_{i=1}^d [\lambda_i f_i(\boldsymbol{w}) + (c_i - \lambda_i)g_i(\boldsymbol{w})] + h(\boldsymbol{w}).$$

In this section, we will prove the following:

$$\mathrm{sign}\,(g_{i*}(\boldsymbol{w}) - f_{i*}(\boldsymbol{w}))(\nabla f_{i*}(\boldsymbol{w}) - \nabla g_{i*}(\boldsymbol{w}))^\top \boldsymbol{H_\lambda}^{-1}(\nabla f_i(\boldsymbol{w}) - \nabla g_i(\boldsymbol{w})) \in \partial_{\lambda_i} F(\boldsymbol{\lambda}), \ \forall i.$$

Assume that $\sum_{i=1}^d [\lambda_i \nabla^2 f_i(\boldsymbol{w_\lambda}) + (c_i - \lambda_i)\nabla^2 g_i(\boldsymbol{w_\lambda})] + \nabla^2 h(\boldsymbol{w_\lambda}) \succ 0$ for every $\boldsymbol{\lambda} \in \Lambda$. In this part of the proof, we will denote $\boldsymbol{w_\lambda}$ by $\boldsymbol{w}$ for simplicity.

To compute $\frac{\mathrm{d}\boldsymbol{w}}{\mathrm{d}\lambda_i}$, we implicitly differentiate (with respect to $\lambda_i$) the following stationary equation.

$$\sum_{j=1}^{d}[\lambda_j \nabla f_j(\boldsymbol{w}) + (c_j - \lambda_j)\nabla g_j(\boldsymbol{w})] + \nabla h(\boldsymbol{w}) = 0$$

$$\Rightarrow \nabla f_i(\boldsymbol{w}) + \lambda_i \nabla^2 f_i(\boldsymbol{w}) \cdot \frac{\partial \boldsymbol{w}}{\partial \lambda_i} - \nabla g_i(\boldsymbol{w}) + (c_i - \lambda_i)\nabla^2 g_i(\boldsymbol{w}) \cdot \frac{\partial \boldsymbol{w}}{\partial \lambda_i}$$

$$+ \sum_{1 \le j \le d,\ j \ne i} \left[ \lambda_j \nabla^2 f_j(\boldsymbol{w}) \cdot \frac{\partial \boldsymbol{w}}{\partial \lambda_i} + (c_j - \lambda_j)\nabla^2 g_j(\boldsymbol{w}) \cdot \frac{\partial \boldsymbol{w}}{\partial \lambda_i} \right] + \nabla^2 h(\boldsymbol{w}) \cdot \frac{\partial \boldsymbol{w}}{\partial \lambda_i} = 0$$

By rearranging terms, we have

$$\left( \sum_{j=1}^{d} \left[ \lambda_j \nabla^2 f_j(\boldsymbol{w}) + (c_j - \lambda_j)\nabla^2 g_j(\boldsymbol{w}) \right] + \nabla^2 h(\boldsymbol{w}) \right) \frac{\partial \boldsymbol{w}}{\partial \lambda_i} = -(\nabla f_i(\boldsymbol{w}) - \nabla g_i(\boldsymbol{w})).$$

By the assumption, $\boldsymbol{H}_{\boldsymbol{\lambda}} := \sum_{j=1}^{d} \left[ \lambda_j \nabla^2 f_j(\boldsymbol{w}) + (c_j - \lambda_j)\nabla^2 g_j(\boldsymbol{w}) \right] + \nabla^2 h(\boldsymbol{w})$ is positive definite and hence invertible. Thus,

$$\frac{\partial \boldsymbol{w}}{\partial \lambda_i} = -\boldsymbol{H}_{\boldsymbol{\lambda}}^{-1}(\nabla f_i(\boldsymbol{w}) - \nabla g_i(\boldsymbol{w})).$$

Now observe that $F(\boldsymbol{\lambda}) = |f_{i^*}(\boldsymbol{w}_{\boldsymbol{\lambda}}) - g_{i^*}(\boldsymbol{w}_{\boldsymbol{\lambda}})|$. Therefore,

$$\mathrm{sign}\,(f_{i^*}(\boldsymbol{w}) - g_{i^*}(\boldsymbol{w}))\frac{\partial}{\partial \lambda_i}(f_{i^*}(\boldsymbol{w}) - g_{i^*}(\boldsymbol{w})) \in \partial_{\lambda_i} F(\boldsymbol{\lambda}).$$

Since

$$\frac{\partial}{\partial \lambda_i}(f_{i^*}(\boldsymbol{w}) - g_{i^*}(\boldsymbol{w})) = -(\nabla f_{i^*}(\boldsymbol{w}) - \nabla g_{i^*}(\boldsymbol{w}))^{\top} \boldsymbol{H}_{\boldsymbol{\lambda}}^{-1}(\nabla f_i(\boldsymbol{w}) - \nabla g_i(\boldsymbol{w})),$$

we have

$$-\,\mathrm{sign}\,(f_{i^*}(\boldsymbol{w}) - g_{i^*}(\boldsymbol{w}))(\nabla f_{i^*}(\boldsymbol{w}) - \nabla g_{i^*}(\boldsymbol{w}))^{\top} \boldsymbol{H}_{\boldsymbol{\lambda}}^{-1}(\nabla f_i(\boldsymbol{w}) - \nabla g_i(\boldsymbol{w})) \in \partial_{\lambda_i} F(\boldsymbol{\lambda}). \quad (4)$$

### A.5 RATIONALE AND INTUITION BEHIND THE APPROXIMATION

We continue from Sec. 3.2 and provide more justifications for the gradient approximation technique. Assume that $\sum_{i=1}^{d}[\lambda_i \nabla^2 f_i(\boldsymbol{w}_{\boldsymbol{\lambda}}) + (c_i - \lambda_i)\nabla^2 g_i(\boldsymbol{w}_{\boldsymbol{\lambda}})] + \nabla^2 h(\boldsymbol{w}_{\boldsymbol{\lambda}}) \succ 0$ for every $\boldsymbol{\lambda} \in \Lambda$. Then, the gradient can be fully characterized as in equation 4.

The rationale behind the approximation $\boldsymbol{\gamma} \approx (0, 0, \dots, \gamma_{i^*}, \dots, 0)$ is that $|\gamma_{i^*}|$ will be maximized at $i^*$ if $\|\nabla f_1(\boldsymbol{w}) - \nabla g_1(\boldsymbol{w})\| \approx \|\nabla f_2(\boldsymbol{w}) - \nabla g_2(\boldsymbol{w})\| \approx \cdots \approx \|\nabla f_d(\boldsymbol{w}) - \nabla g_d(\boldsymbol{w})\|$. This is because $(\nabla f_{i^*}(\boldsymbol{w}) - \nabla g_{i^*}(\boldsymbol{w}))^{\top} \boldsymbol{H}_{\boldsymbol{\lambda}}^{-1}(\nabla f_i(\boldsymbol{w}) - \nabla g_i(\boldsymbol{w}))$ is an inner product between $\boldsymbol{H}_{\boldsymbol{\lambda}}^{-1/2}(\nabla f_{i^*}(\boldsymbol{w}) - \nabla g_{i^*}(\boldsymbol{w}))$ and $\boldsymbol{H}_{\boldsymbol{\lambda}}^{-1/2}(\nabla f_i(\boldsymbol{w}) - \nabla g_i(\boldsymbol{w}))$, and they are always perfectly aligned when $i = i^*$.

This approximation is also intuitive. Recall that changing $\lambda_{i^*}$ affects the weights associated with $f_{i^*}(\boldsymbol{w})$ and $g_{i^*}(\boldsymbol{w})$ in the inner optimization problem. Thus, changes in $\lambda_{i^*}$ will directly affect $F(\boldsymbol{\lambda}) = |f_{i^*}(\boldsymbol{w}) - g_{i^*}(\boldsymbol{w})|$. On the other hand, changing $\lambda_i$ for $i \ne i^*$ does not affect the weights associated with $f_{i^*}(\boldsymbol{w})$ and $g_{i^*}(\boldsymbol{w})$ but only affects the weights of other terms, so it will only indirectly and weakly affect $F(\boldsymbol{\lambda})$.

### A.6 FAIRBATCH ALGORITHMS IN PSEUDOCODE

We continue from Sec. 3.2 and present the FairBatch algorithms in pseudocode. Algorithms 2, 3, and 4 show how $\boldsymbol{\lambda}$ is adjusted for equal opportunity, equalized odds, and demographic parity, respectively. From the intermediate model at each epoch (or after a certain iterations), we first obtain $f(\boldsymbol{w})$ and $g(\boldsymbol{w})$, which correspond to the losses conditioned on each class. Then, one can update the current value of $\boldsymbol{\lambda}$ by comparing $f(\boldsymbol{w})$ and $g(\boldsymbol{w})$.

---

**Algorithm 2:** Adaptive adjustment of $\lambda$ w.r.t. equal opportunity.

---

**Input:** Intermediate model, criterion, train data $(x_{train}, z_{train}, y_{train})$,
previous lambda $\lambda^{(t-1)}$, and FairBatch's learning rate $\alpha$

output = model $(x_{train})$

$loss$ = criterion (output, $y_{train}$)

$$\lambda^{(t)} = \begin{cases} \lambda^{(t-1)} + \alpha, & \text{if } \text{mean}(loss[(\text{y}=1, \text{z}=0)]) > \text{mean}(loss[(\text{y}=1, \text{z}=1)]) \\ \lambda^{(t-1)} - \alpha, & \text{if } \text{mean}(loss[(\text{y}=1, \text{z}=0)]) < \text{mean}(loss[(\text{y}=1, \text{z}=1)]) \\ \lambda^{(t-1)}, & \text{otherwise} \end{cases}$$

**Output :** Next lambda $\lambda^{(t)}$

---

**Algorithm 3:** Adaptive adjustment of $\boldsymbol{\lambda}$ w.r.t. equalized odds.

---

**Input:** Intermediate model, criterion, train data $(x_{train}, z_{train}, y_{train})$,
previous lambda $\boldsymbol{\lambda}^{(t-1)}$, and FairBatch's learning rate $\alpha$

output = model $(x_{train})$

$loss$ = criterion (output, $y_{train}$)

$d_{\text{y}=0}$ = mean$(loss[(\text{y}=0, \text{z}=0)])$ − mean$(loss[(\text{y}=0, \text{z}=1)])$

$d_{\text{y}=1}$ = mean$(loss[(\text{y}=1, \text{z}=0)])$ − mean$(loss[(\text{y}=1, \text{z}=1)])$

**if** $|d_{\text{y}=0}| > |d_{\text{y}=1}|$ **then**

$$\lambda_1^{(t)} = \begin{cases} \lambda_1^{(t-1)} + \alpha, & \text{if } d_{\text{y}=0} > 0 \\ \lambda_1^{(t-1)} - \alpha, & \text{if } d_{\text{y}=0} < 0 \\ \lambda_1^{(t-1)}, & \text{otherwise} \end{cases}$$

**else**

$$\lambda_2^{(t)} = \begin{cases} \lambda_2^{(t-1)} + \alpha, & \text{if } d_{\text{y}=1} > 0 \\ \lambda_2^{(t-1)} - \alpha, & \text{if } d_{\text{y}=1} < 0 \\ \lambda_2^{(t-1)}, & \text{otherwise} \end{cases}$$

**Output :** Next lambda $\boldsymbol{\lambda}^{(t)}$

---

## B APPENDIX – EXPERIMENTS

### B.1 OTHER EXPERIMENTAL SETTINGS

We continue from Sec. 4 and provide more details on experimental settings. We use the Adam optimizer for all trainings. We perform cross-validation on the training sets to find the best hyper-parameters for each algorithm. We evaluate models on separate test sets, and the ratios of the train versus test data for the synthetic and real datasets are 2:1 and 4:1, respectively.

### B.2 EQUALIZED ODDS RESULTS

We continue from Sec. 4.1 and show Table 4, which compares the performances of all the fair training techniques on the synthetic, COMPAS, and AdultCensus test sets w.r.t. equalized odds. The key observations are the same as in Table 1 where overall FairBatch has the most robust performance against the state of the arts w.r.t. accuracy, fairness, and runtime.

### B.3 EXTENSION OF ADAFAIR

We continue from Sec. 4 and provide more details on how we extend AdaFair, which already supports ED, to also support EO and DP. The extension to EO is straightforward as EO is a relaxed version of ED where only the y = 1 class is considered when measuring disparity. Hence, we only reweight examples in the y = 1 class as well. The extension to DP is done by giving more weights on the

---

**Algorithm 4:** Adaptive adjustment of $\boldsymbol{\lambda}$ w.r.t. demographic parity.

---

**Input:** Intermediate model, criterion, train data $(x_{train}, z_{train}, y_{train})$,
      previous lambda $\boldsymbol{\lambda}^{(t-1)}$, and FairBatch's learning rate $\alpha$

output = model $(x_{train})$

$loss$ = criterion (output, $\mathbf{1}$)

$d_{\text{y}=0}$ = sum($loss[(\text{y}=0, \text{z}=0)]$)/len(z = 0) − sum($loss[(\text{y}=0, \text{z}=1)]$)/len(z = 1)

$d_{\text{y}=1}$ = sum($loss[(\text{y}=1, \text{z}=0)]$)/len(z = 0) − sum($loss[(\text{y}=1, \text{z}=1)]$)/len(z = 1)

**if** $|d_{\text{y}=0}| > |d_{\text{y}=1}|$ **then**

$$\lambda_1^{(t)} = \begin{cases} \lambda_1^{(t-1)} - \alpha, & \text{if } d_{\text{y}=0} > 0 \\ \lambda_1^{(t-1)} + \alpha, & \text{if } d_{\text{y}=0} < 0 \\ \lambda_1^{(t-1)}, & \text{otherwise} \end{cases}$$

**else**

$$\lambda_2^{(t)} = \begin{cases} \lambda_2^{(t-1)} + \alpha, & \text{if } d_{\text{y}=1} > 0 \\ \lambda_2^{(t-1)} - \alpha, & \text{if } d_{\text{y}=1} < 0 \\ \lambda_2^{(t-1)}, & \text{otherwise} \end{cases}$$

**Output :** Next lambda $\boldsymbol{\lambda}^{(t)}$

---

Table 4: Performances on the synthetic, COMPAS, and AdultCensus test sets w.r.t. equalized odds (ED). The other settings are identical to Table 1.

| | Synthetic | | | COMPAS | | | AdultCensus | | |
|---|---|---|---|---|---|---|---|---|---|
| Method | Acc. | ED Disp. | Epochs | Acc. | ED Disp. | Epochs | Acc. | ED Disp. | Epochs |
| LR | .885±.000 | .115±.000 | 400 | .681±.002 | .239±.006 | 300 | .845±.001 | .056±.003 | 300 |
| Cutting | .859±.001 | .036±.004 | 650 | .665±.005 | .066±.018 | 400 | .802±.001 | .062±.005 | 600 |
| RW | .856±.000 | .038±.002 | 350 | .685±.000 | .137±.000 | 300 | .835±.001 | .134±.006 | 100 |
| LBC | .858±.001 | **.026±.000** | 8800 | .673±.002 | .063±.005 | 9000 | .840±.002 | **.027±.004** | 3300 |
| FC | .865±.000 | .030±.001 | 800 | .677±.004 | .101±.024 | 50 | .841±.001 | .038±.003 | 300 |
| AD | .857±.000 | .030±.001 | 1200 | .683±.000 | .082±.027 | 450 | .843±.002 | .033±.002 | 500 |
| AdaFair | .868±.001 | .029±.002 | 22400 | .675±.000 | .066±.002 | 9600 | .843±.001 | .038±.002 | 7800 |
| **FairBatch** | .856±.001 | .038±.002 | 400 | .682±.001 | **.052±.014** | 100 | .843±.001 | .036±.002 | 500 |

positive examples of a certain sensitive group z = $z$ that suffers from a lower positive prediction rate than other groups.

## B.4 FAIRNESS CURVES

We continue from Sec. 4.1 and show in Figures 3 and 4 the EO and DP disparity curves against the number of epochs for each fairness technique on the synthetic dataset. We also directly compare the curves of all fairness techniques in one graph as shown in Figure 5. Since LBC and AdaFair require more than 10x many epochs than other methods, we only show their first 1000 epochs. As a result, FairBatch is one of the fastest methods to converge to low EO or DP disparities.

## B.5 TRADE-OFF CURVES OF FAIRBATCH

We continue from Sec. 4.1 and show in Fig. 6 the accuracy-fairness trade-off curves of FairBatch for EO and DP on the synthetic dataset. FairBatch can be tuned by making it "less sensitive" to disparity. In Algorithms 2 and 4, notice that the $\lambda$ parameters are updated if there is any disparity among sensitive groups. We modify this logic where the $\lambda$ parameters are only updated if the disparity is

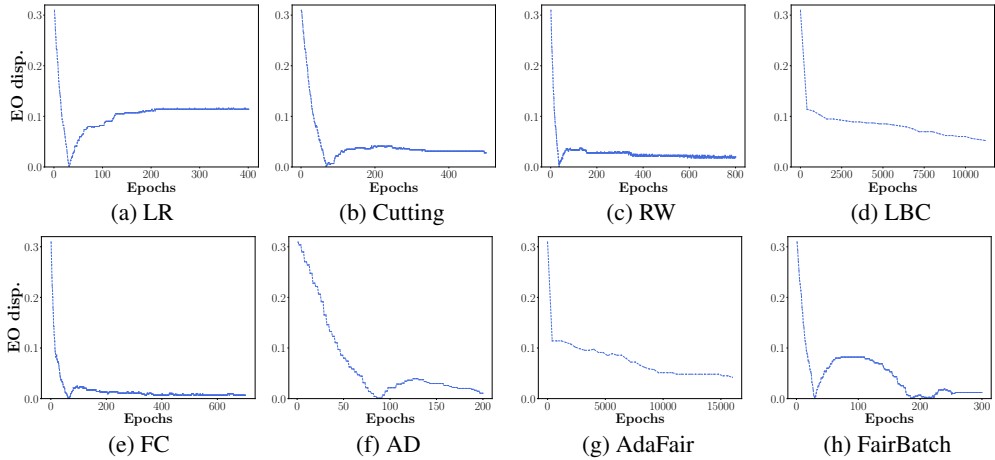

Figure 3: EO disparity curves of algorithms on the synthetic dataset.

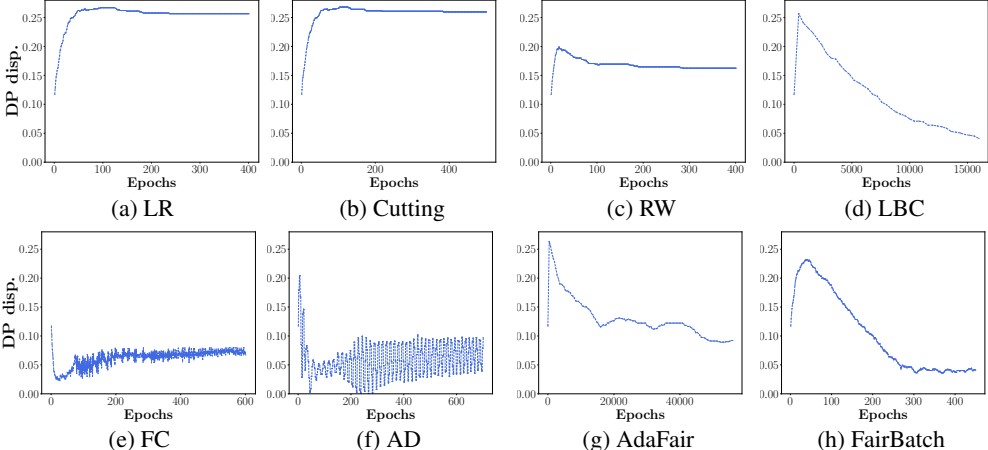

Figure 4: DP disparity curves of algorithms on the synthetic dataset.

above some threshold $T$. The trade-off curves in Fig. 6 are thus generated by adjusting $T$. For both EO and DP, we observe that there is a clear trade-off between accuracy and disparity.

### B.6 WALL CLOCK TIMES

We continue from Sec. 4.1 and show in Table 5 the wall clock times (in seconds) of the experiments in Table 1 where we compare FairBatch against all the fairness techniques on the synthetic, COMPAS, and AdultCensus datasets. As a result, each runtime is proportional to the number of epochs shown in Table 1. When comparing the runtimes of individual batches, FairBatch's batch takes 1.5x longer to run than LR's batch.

Table 5: Wall clock times (in seconds) of the experiments in Table 1 using the same settings.

| Dataset | LR | Cutting | RW | LBC | FC | AD | AdaFair | FairBatch |
|---|---|---|---|---|---|---|---|---|
| Synthetic | 5.71 | 5.67 | 17.24 | 208.47 | 16.05 | 3.97 | 294.31 | 5.25 |
| COMPAS | 6.07 | 3.34 | 7.48 | 94.10 | 2.76 | 6.93 | 215.39 | 3.00 |
| AdultCensus | 22.96 | 7.70 | 10.02 | 558.31 | 28.71 | 31.76 | 791.58 | 46.79 |

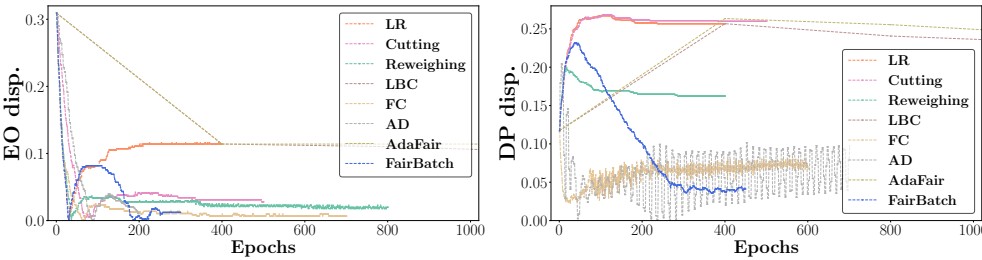

(a) EO disparity curve of FairBatch and the baselines. (b) DP disparity curve of FairBatch and the baselines.

Figure 5: Epochs-fairness disparity curves of all algorithms together.

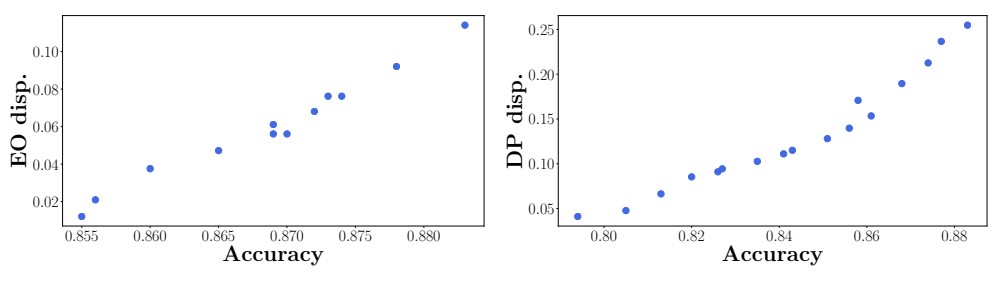

(a) Accuracy-EO disparity trade-off curve.  (b) Accuracy-DP disparity trade-off curve.

Figure 6: Accuracy-fairness disparity trade-off curves of FairBatch on the synthetic dataset.

## B.7 COMPARISON WITH ADAFAIR

We continue from Sec. 4.1 and compare the class weights between FairBatch and the AdaFair algorithm. For AdaFair, the class weights are calculated by adding all example weights in each class. Fig. 7 shows the weight changes of each algorithm. Overall, the trends of the weights are similar. Again, the advantage of FairBatch is that it can run within one model training instead of using multiple model trainings as in AdaFair.

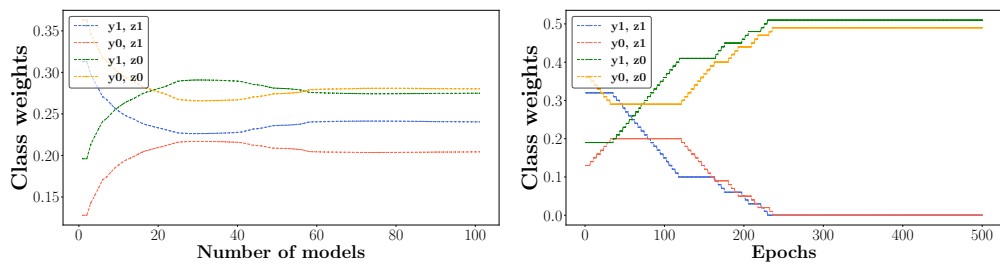

(a) Number of models-class weight curve of AdaFair.  (b) Epochs-class weight curve of FairBatch.

Figure 7: Comparison of the weight changes on AdaFair and FairBatch w.r.t. equalized odds on the synthetic dataset.

## B.8 EXTENSION OF FAIRBATCH TO MULTI CLASSIFICATION

We continue from Sec. 4.2 and explain how FairBatch can be extended to support multi classification by adjusting more $\lambda$ parameters. For example, for ED, the label attribute has $n$ classes, and each class connects to $m$ $\lambda$s. We adjust $m$ $\lambda$s in the class $y = i$ at each epoch, where the class $y = i$ has the highest ED disparity at that epoch.

### B.9    FAIRBATCH WITH IMPORTANCE SAMPLING

We continue from Sec. 4.3 and show Fig. 8, which plots the convergence of FairBatch when merged with loss-based weighting batch selection. As a result, FairBatch uses about 50 fewer epochs to converge to low disparities compared to not using loss-based weighting.

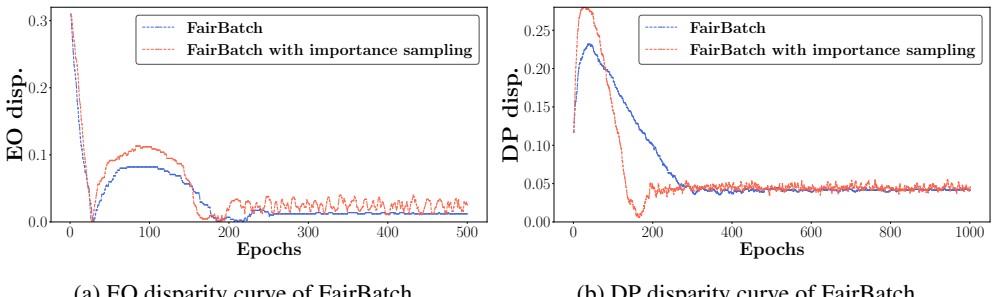

(a) EO disparity curve of FairBatch.          (b) DP disparity curve of FairBatch.

Figure 8: Fairness curves of FairBatch on the synthetic dataset, with/without loss-based weighting (Loshchilov and Hutter, 2016).

