# OpenReview forum: "FairBatch: Batch Selection for Model Fairness"
_ICLR.cc/2021/Conference — ICLR 2021 Poster_

### Official Review · AnonReviewer3 · 2020-10-28
**FairBatch: Batch Selection for Model Fairness**

**Rating:** 4
**Confidence:** 4

**Review:**

In this paper, the authors study the problem of training fair machine learning models through the lens of bi-level optimization. In particular, they propose a method, denoted FairBatch, that adaptively selects different batch-sizes for different protected groups to impose a certain measure of fairness. This is achieved by solving an outer optimization problem that imposes fairness by computing a batch-size ratio for different groups. This batch selection is then used to train the original model. The proposed approach can impose several prominent fairness measures, among which: equal opportunity, equalized odds, and demographic parity.  To demonstrate the efficiency of FairBatch, the authors conducted several experiments on synthetic and real datasets. The results show the proposed method is easily implementable, scalable and can achieve comparable performance when compared to existing methods.

Pros:
The authors approached the problem of fair machine learning by iteratively adapting the batch-size ratio for different protected groups. The approach is novel and was implemented through a novel bi-level optimization formulation.

The proposed method is easy to implement, scalable, and can readily improve fairness for any pre-trained model.

Clarity: The main body of the paper is concise and clearly written with very few typos.

Cons:
The theoretical part of the paper requires a more careful and detailed analysis. First, the inner minimization problem in section 3.1 has $w_{\lambda}$ appearing in the objective. I think this should be $w$ instead. This also affects the proof of Lemma 2 in the supplementary material. Moreover, the author computes the gradient of the objective of the outer optimization problem, $F(\lambda)$. Being a finite max, this function might not be differentiable.

It seems that positive definiteness assumptions in sections A.1 and A.4 are not easy to check. The authors did not comment on the practicality of these assumption nor they provided a method to check whether they hold. Moreover, I believe these assumptions should be moved to the main body of the paper.

I believe the paper is worth publishing after re-working the theoretical part.

Minor Comments:
1.	Page 3: In defining the bi-level optimization problem, the objective function of the inner minimization is a function of $w$. This should be explicitly included in the formulation (it is currently hidden in $\hat{y_i}$)
2.	In the proof of proposition 2 in Appendix A, should it be $\ell(|1-y|, \cdot) = 1 - \ell(y,\cdot)$?
3.	In Sections A.2 and A.4, the subscript of the functions $f_i$ and $g_i$ are missing in some expressions (also in Lemma 1).

The related material are referenced and well-discussed in the paper. The authors clearly positioned their work in the related field and discussed their contributions in comparison to other similar works.

-------------------------------------------------------------------------------------------------------------------------------------------------------------------------------
Update after the author's response:

I was not convinced by many of the authors' responses.

1. While the author(s) agree that $F(\lambda)$ is non-differentiable, they keep the gradient updates in the paper with a footnote referring to sub-gradient methods in (Boyd et al. 2004)? Moreover, it is not clear what the sub-gradient would refer too when the loss function is non-convex, what would $H_{\lambda}$ refer to in section 3.2? One possible way to solve this issue might be through a mild smoothing technique. But as presented the theoretical part is not rigorous.

2. The positive definiteness assumption in Lemma 1, although will most probably hold when the three functions are convex; this limits the application of the results as typical ML loss functions involve non-convexity.

3. My comment on the subscript of the functions $f_i$ and $g_i$ in Lemma 1 was addresses by adding a comment mentioning that the subscript will removed for the simpler notation. I believe one subscript does not make the notation complex. Rather defining two notations might confuse the reader.

Despite my tendency to have this novel idea and work published, I believe the authors need to be more careful in writing and dealing with the theoretical section of the paper. I will hence decrease the score to 4.

---

> ### Author Response · Authors · 2020-11-17
> **Response to Reviewer 3**
>
> We thank you for the insightful comments.
>
> Q3-1. Theoretical part
>
> A3-1.
>
> (1) Inner minimization problem, Section 3.1
>
> Thanks for pointing out the typo. We now fixed it -- see Sections 3.1 and A.2, highlighted in blue. Regarding the proof of Lemma 2 (now part of Lemma 1), we clarify that we intentionally substituted $w$ of the inner optimization with $w_\lambda$ to prove the quasi-convexity of $F(\lambda)$.
>
> (2) Gradient of $F(\lambda)$
>
> That’s right. $F(\lambda)$ may not be differentiable on corner points, so we use subgradient methods (Boyd et al., 2004) that work on non-differentiable functions. We clarified this point in our revision (Section 3.2, footnote, highlighted in blue).
>
>
> Q3-2. Positive definiteness assumptions
>
> A3-2.
> Here we echo the positive definiteness assumption in A.2 (we think you mean A.2 not A.1) and A.4: $\lambda \nabla^2 f_1(w_\lambda) + (c_1-{\lambda}) \nabla^2 g_1(w_\lambda) + \nabla^2 h(w_\lambda) \succ 0$. For instance, if $f_1$, $g_1$, and $h$ are convex, this condition will hold unless all the three functions share their stationary points, which is very unusual. We now added this argument and moved them to the main body in our revision (Section 3.1, highlighted in blue).
>
>
> Q3-3. Minor comments
>
> A3-3. We reflected all your comments in our revision (Sections 2, 3.1, A.1, A.2, and A.4, highlighted in blue).

---

### Official Review · AnonReviewer4 · 2020-10-28
**Review of FairBatch: Batch Selection for Model Fairness**

**Rating:** 7
**Confidence:** 4

**Review:**

The paper addresses the problem of fairness by viewing the task of learning as a bilevel optimization problem. The task of learning is decomposed to two levels: 1- Choosing a batch of data samples so as to satisfy a given fairness criterion; 2- training the model using the chosen batch. The main contribution of the paper is that it proposes a way to sample batches of data that would satisfy a given fairness criterion and incorporates that into model training. The authors show the performance of models using their proposed approach on synthetic and real-world data.

The paper is well-written and points to a promising direction in handling fairness using (non-causal) notions of fairness. The point is to find batches of data that mitigate disparities in prediction. The batches are chosen by finding lambdas that minimize the disparity in prediction as opposed to necessarily setting them to zero. This is a fine distinction. The update rules of the resulting optimization problem are also presented.

The proposed approach is tested on multiple datasets. According to Tables 1 and 2, FairBatch does not necessarily improve performance. In some cases, the performance is lower than that of other methods, but given that overall, the performance is comparable with other methods and FairBatch is sometimes also faster in number of epochs, one can say that the potential in FairBatch is worthwhile.

Minor comments:

In Figures 3 and 4 of Section B4, it would be better if we can see the convergence rates plotted in one figure, using different colors for each algorithm. This way, one can compare the results more efficiently.

---

> ### Author Response · Authors · 2020-11-17
> **Response to Reviewer 4**
>
> We do appreciate your comments. As per your great suggestion, we now included a figure that contains all the convergence rates in our revision -- see Section B.4, Figure 5, highlighted in blue.

---

### Official Review · AnonReviewer2 · 2020-10-28
**Nice paper; question about sampling versus weighting**

**Rating:** 6
**Confidence:** 3

**Review:**

This work proposes a modification to stochastic gradient descent with the goal of having the resulting model satisfy fairness constraints. The modification entails changing the sampling procedure to select the minibatches. Rather than sampling the instances at random, the authors propose a procedure which first generates probabilities for each of the instances by first solving an optimization problem which corresponds to different fairness objectives. Training is otherwise carried out as usual. The authors outline the optimizations corresponding to several fairness objectives, and provide an empirical demonstration which compares the proposed approach to the current state of the art where the proposed method performs quite well.

Overall, I think this is a nice, simple, idea which appears to work well in practice. The authors do a commendable job of describing both the problem and solution in plain terms while also providing technical details of the implicit weighting procedure.

What I don’t quite understand is the advantage of modifying the batches versus incorporating the inclusion weights in the loss function itself. It seems that the proposed model will be equivalent to weighting the loss as the number of mini batches grows very large, however the experimental results seem to indicate otherwise. Can the authors provide some intuition for this? Specifically, should we expect equivalent performance between the proposed method and example reweighting if the same probability weights are used as proposed in the paper and we weight random examples by those probabilities as the number of minibatches grows? If so, how should we be thinking about this tradeoff in the case where we use a small to moderate number of minibatches?

Small edits:
- In the first sentence of 3.1 there is a misspelling of optimization.

---

> ### Author Response · Authors · 2020-11-17
> **Response to Reviewer 2**
>
> We thank you for the great comments.
>
> Q2-1. Relationship between FairBatch and loss weighting
>
> A2-1.
> You are right. FairBatch and loss reweighting would indeed converge to the same result. This equivalence has been discussed in other fairness literature as well (Agarwal et al., 2018; Jiang et al., 2020), and we will also clarify this in our revision. A key distinction is in an implementation aspect -- FairBatch is easy to be incorporated in any current system, requiring only a single line of change in code.
>
> We clarify why the experimental results seem to indicate otherwise. The only methods that use example reweighting are RW (Kamiran and Calders, 2011) and LBC (Jiang et al., 2020). However, these methods are not the weighted loss versions of FairBatch, but completely different techniques. Specifically, RW assigns weights to examples to reduce discrimination, but then sticks with these weights throughout the entire model training, unlike FairBatch. LBC iteratively reweights examples, but with the strong assumption that there exists a data distribution without any bias. We added a clarification in our revision (Section 4, Baselines, highlighted in blue).
>
> If we were to compare FairBatch with its weighted loss version, we do expect equivalent performance, especially as the number of minibatches grows.
>
>
> Q2-2. Small edits
>
> A2-1. We now fixed the typo.

---

### Official Review · AnonReviewer1 · 2020-10-30
**Adaptively select mini-batch for sensitive groups to improve model fairness**

**Rating:** 6
**Confidence:** 5

**Review:**

The problem is well-motivated. The paper is well-organized and written. The claims are well-supported by theoretical analysis and experimental results. In contrast with previous work that requires nontrivial re-configurations in machine learning, FairBatch formalizes the sampling probability as an implicit connection between the inner (for fairness criterion) and outer (task) optimizer in bilevel optimization.
[Pro 1] This paper provides insights into fairness and machine learning from the lens of bilevel optimization, emphasizing the inverse proportional relationship between sampling probability and loss for sensitive groups.
[Pro 2] FairBatch is not only consistent with intuition but also easy to implement.
[Con 1] It would be helpful for the authors to summarize their contributions more from the fairness aspect. I am confused about the ability of FairBatch to mitigate disparate accuracy. Is it designed especially for the unfairness caused by minimizing average error that fits majority populations [to clarify my point, please refer to the 2nd cause in Section 2.1, abs-1810-08810]?
[abs-1810-08810] Alexandra Chouldechova, Aaron Roth: The Frontiers of Fairness in Machine Learning. CoRR abs/1810.08810 (2018)
[Con 2] Is the number of total disparities (as the authors claimed in Section 3.2) d in Section 3 the lambda dimension? No explanation is provided for this notion. If true, is d equal to n_z-1? Why not C(n_z, 2)? (C stands for Combination). In practice, when the sensitive space is sometimes continuous and big, there would be too many pairs to compute. More details about d will strengthen the submission. I am not requesting more experiments, but want to understand how FairBatch works in practice.
[Con 3] As for the experiments, the analysis fails to go beyond single-dimensional performance summaries. Also, alpha is a vital hyper-parameter. However, the criterion used for selecting the final setting is not clear in this paper.

---

> ### Author Response · Authors · 2020-11-17
> **Response to Reviewer 1**
>
> We thank you for the insightful comments.
>
> Q1-1. Contributions in fairness
>
> A1-1.
> FairBatch has two key contributions from the fairness aspect. One is to adjust the sensitive group ratios for fairness; see below for details. Another is making it very easy to improve the fairness of any ML system, as the reviewer acknowledges.
>
> We elaborate how FairBatch mitigates the three unfairness causes described in (Chouldechova et al., 2018): (A) minimizing average error fits majority populations, (B) bias encoded in data, and (C) the need to explore and gather more data. As you suspected, FairBatch addresses the cause (A) via balancing the sensitive group ratios within a batch. FairBatch also addresses (B) in some cases. For example, consider the recidivism prediction problem described in (Chouldechova et al., 2018) where minority populations have biased labels. In this case, FairBatch can be configured to make the recidivism prediction rate for the minority population similar to those of other populations. There may be other types of data bias that FairBatch is not able to address. Finally, FairBatch does not directly address (C) where one must gather more data for better fairness. Instead, there is a recent line of work that studies data collection techniques for fairness.
>
> We will summarize these points in our revision.
>
>
> Q1-2. Details for d
>
> A1-2.
> You are right. The total number of disparities d is indeed the same as the lambda dimension. FairBatch has one lambda per disparity objective. As you suspected, d should read C(n_z, 2). In our implementation, however, we only use d = n_z-1 disparity objectives as an approximation for better efficiency. We clarified these points with justification in our revision (Sections 2 and 3, highlighted in blue).
>
>
> Q1-3. Experiments
>
> A1-3.
>
> (1) FairBatch for multiple dimensions
>
> We would like to clarify that both equalized odds and demographic parity have d = 2, so all experiments involving these measures (Tables 2, 3, and 4) are multi-dimensional results. In page 4, notice that both measures have *two disparities*. We clarified these points in our revision (Section 4, Measuring Fairness, highlighted in blue).
>
> (2) Selecting alpha
>
> We start from a candidate set of values within the range [0.0001, 0.05] and use cross-validation on the training set to choose the value that results in the highest accuracy with low fairness violation. We added these details in our revision (Section 4, FairBatch settings, highlighted in blue).

---

### Decision · Program_Chairs · 2021-01-07
**Final Decision**

**Decision:**

Accept (Poster)

**Comment:**

All the reviewers and I agree that the proposed approach is interesting and the paper is overall well written. However, I agree with R3 that the paper  need further re-working the theoretical part (see the post-rebuttal comments of R4). Thus, I would encourage the authors to carefully address the comments of the reviewers in the revised version of the paper, which would ultimately improve the quality of the paper.